# Ethanol facilitates socially evoked memory recall in mice by recruiting pain-sensitive anterior cingulate cortical neurons

Tetsuya Sakaguchi[1], Satoshi Iwasaki[1], Mami Okada[1], Kazuki Okamoto[1] & Yuji Ikegaya [1,2]

Alcohol is a traditional social-bonding reinforcer; however, the neural mechanism underlying ethanol-driven social behaviors remains elusive. Here, we report that ethanol facilitates observational fear response. Observer mice exhibited stronger defensive immobility while observing cagemates that received repetitive foot shocks if the observer mice had experienced a brief priming foot shock. This enhancement was associated with an observation-induced recruitment of subsets of anterior cingulate cortex (ACC) neurons in the observer mouse that were responsive to its own pain. The vicariously activated ACC neurons projected their axons preferentially to the basolateral amygdala. Ethanol shifted the ACC neuronal balance toward inhibition, facilitated the preferential ACC neuronal recruitment during observation, and enhanced observational fear response, independent of an oxytocin signaling pathway. Furthermore, ethanol enhanced socially evoked fear response in autism model mice.

[1] Graduate School of Pharmaceutical Sciences, The University of Tokyo, Tokyo 113-0033, Japan. [2] Center for Information and Neural Networks, National Institute of Information and Communications Technology, Suita City, Osaka 565-0871, Japan. Correspondence and requests for materials should be addressed to Y.I. (email: yuji@ikegaya.jp)

Alcohol has been enjoyed throughout the history of humankind, as it promotes sociality[1]. Acute alcohol consumption reduces social stress[2], increases generosity[3], and improves recognition of facial expressions[4]. Recent studies have demonstrated that it enhances social bonding[5] and promotes smile contagion[6], suggesting that alcohol enhances affective empathy (or emotional contagion), which underlies altruistic motivation and prosocial behaviors. However, the neural mechanism of alcohol-driven sociality remains to be identified.

Sociality is enhanced by prior similar experiences. For example, prior similar experiences of the events in stories increase empathy for the persons in the stories[7]. Preis et al.[8] reported that prior pain exposures facilitate subsequent state empathy for pain. Human brain imaging studies suggest that vicarious experience of others' pain involves shared representation of experienced and observed pain[9,10]. This notion raises a possibility that this overlapping neural representation may underlie the experience-dependent facilitation of empathy. However, voxel-level analyses using functional magnetic resonance imaging do not necessarily support the shared representation hypothesis at the single-neuron level[11].

Empathy is a high-level affective process that is often expressed by humans, but rodents also manifest some aspects of empathy-like behaviors[12–14]. Affective empathy is modeled using the fear observational system, in which an animal (observer) exhibits behavioral defensive immobility when it observes the distress of a conspecific (demonstrator) receiving electrical shocks[15,16]. In this fear transmission system, the anterior cingulate cortex (ACC), a pain-relevant brain region, is critical for experiencing vicarious pain[15], consistent with human imaging studies[9,10]. However, fundamental questions remain unsolved, including (i) whether firsthand and vicarious pain activate the same ACC neurons at the single-cell level and (ii) if so, whether ethanol modulates the overlapping ACC neuronal representation and observational fear.

Like ethanol, the neuropeptide oxytocin is also involved in social functioning, including consolation behavior[14], maternal care[17], sociosexual behavior[18], and social recognition[19]. Furthermore, intranasal oxytocin improves behavioral and neural deficits in autism[20]. Regarding empathy, intranasal oxytocin enhances affective empathy in response to both positive- and negative-valence stimuli[21], and facilitates the perceptions of harm for victims[22]. However, the social effect of oxytocin depends on context. For example, oxytocin does not modulate empathic responses to painful pictures but facilitates them when participants are asked to take the perspective of others[23]. Moreover, the social effect of oxytocin differs between in-group and out-group members[24,25]. In this work, we also examined the involvement of endogenous oxytocin signaling in observational fear transmission and compared it to the effect of exogenous ethanol. As a result, we found that mice treated with ethanol-increased defensive immobility while observing the distress of demonstrators. The effect of ethanol emerged independently of the oxytocin signaling.

## Results

**Ethanol enhances socially evoked fear memory recall**. We first examined the effect of a prior shock experience on subsequent fear observation in mice. A total of 80 pairs of cagemates cohoused for 1–2 weeks were randomly divided into four groups, no priming shock/fear observation (no-PS/FO), FO-only, PS-only, and PS + FO (Fig. 1a), and each pair consisted of a randomly chosen observer and demonstrator. In the PS + FO group, the observers were briefly placed in a shock chamber and given a single priming foot shock. Two hours later, the observers were placed in a fear observational chamber in which demonstrators in the neighboring compartment received repetitive foot shocks

(every 12 s for 4 min). In the FO-only group, observers were placed in the same shock chamber but did not receive a priming shock and were then tested for fear observation. In the PS-only group, observers underwent a priming shock, but the demonstrators did not receive repetitive shocks. During the fear observation period, the defensive immobility of the observers in the PS + FO group was significantly higher than that in the three other groups (Fig. 1b), suggesting that the prior shock experience facilitated observational fear response. When the observers in the PS + FO group were re-exposed to the same context 24 h after the observation, they exhibited defensive immobility (Supplementary Fig. 1), indicating that they learned fear-associated contexts by observing their cagemates' behavior. The observers in the PS-only group showed no apparent defensive immobility, and thus, the priming shock alone was insufficient to induce fear response (Fig. 1b and Supplementary Fig. 1). The facilitatory effect of the priming shock lasted for at least 4 w (Supplementary Fig. 2) and was abolished by intraperitoneal injection of 0.1 mg/kg MK801, an N-methyl-D-aspartate (NMDA) receptor inhibitor, 30 min before the priming shock (Supplementary Fig. 3a, b), but not 30 min before fear observation (Supplementary Fig. 3c). The priming shock could not be replaced with other aversive experiences, such as tail pinches or forced swimming (Supplementary Fig. 4), suggesting that observational fear response is facilitated by common experiences shared between demonstrators and observers[16].

A recent study has reported that acute intranasal treatment with oxytocin enhances observational fear in mice[26]. Therefore, we examined the involvement of endogenous oxytocin in observational fear by preventing the oxytocin signaling in the FO-only group. Intraperitoneal injection of 5 mg/kg L-368,899, an oxytocin receptor antagonist, 30 min before the fear observation reduced the duration of defensive immobility (Supplementary Fig. 5). A similar effect of L-368,899 was produced when observer mice had experienced a priming shock (Fig. 1c).

We investigated the effect of ethanol on observational fear response. We adopted a dose of 1.5 g/kg, based on the blood ethanol concentration observed with daily alcohol consumption in humans[27,28]. Observers that had received intraperitoneal injection of ethanol 30 min before the fear observation (i.e., 90 min after the priming shock) exhibited more defensive immobility than saline-treated controls (Fig. 1c), an effect that was not reduced by L-368,899 (Fig. 1c). Thus, ethanol enhances observational fear response in an oxytocin-independent manner. The facilitatory effect of ethanol on observational fear cannot be explained by modulations of locomotor, anxiety, or pain-related behavior because, consistent with previous reports[29,30], this dose of ethanol facilitated locomotor activity (Supplementary Fig. 6a, b) and tended to decrease anxiety-like behaviors (Supplementary Fig. 6c–f), but did not alter pain sensitivity (Supplementary Fig. 6g, h). Moreover, the effect of ethanol was not likely due to enhanced consolidation or retrieval of a simple context-dependent priming shock memory, given that ethanol prevented contextual fear memory recall (Supplementary Fig. 7), and also because observational fear response was enhanced 24 h after the priming shock when ethanol was administered 30 min before the fear observation but not when ethanol was administered 90 min after the priming shock (Supplementary Fig. 8a, b). Importantly, ethanol failed to enhance fear transmission in the FO-only group (Supplementary Fig. 8c). Thus, we conclude that ethanol facilitates socially evoked recall of observer's own fear memory.

**Firsthand and vicarious pain share neuronal representations**. To examine the neuronal representation underlying observational fear at a cellular resolution, we conducted cellular compartment

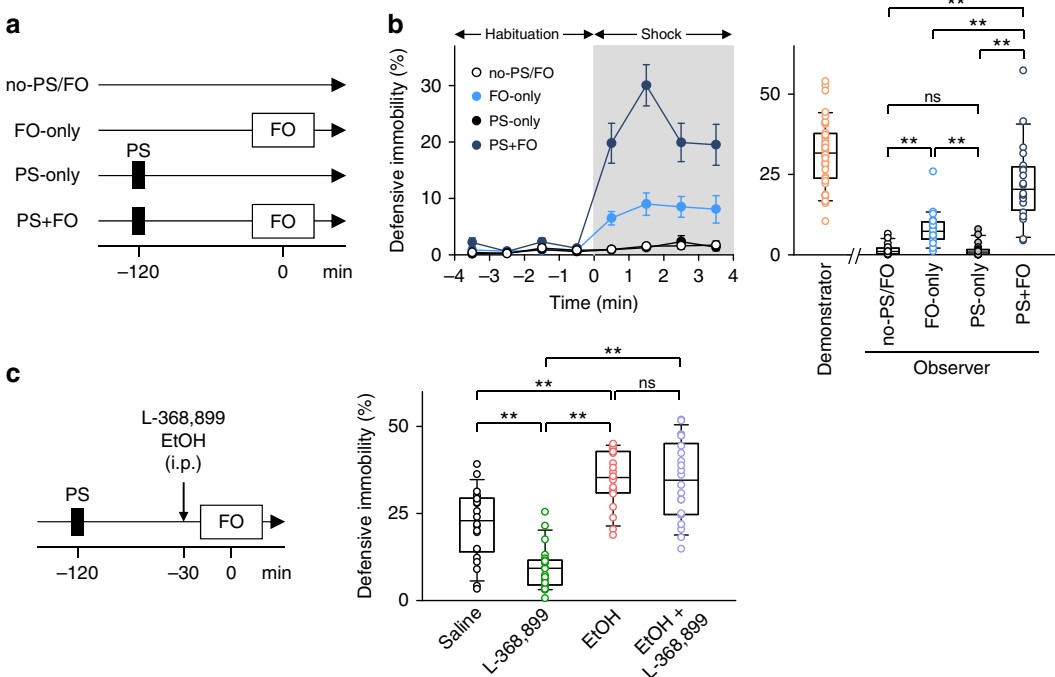

**Fig. 1** Ethanol enhances socially evoked fear memory recall. **a** Experimental paradigm for the four groups in a fear observation (FO) test with a single priming foot shock (PS). **b** *Left*, time course in percentage of time observers exhibited defensive immobility. Repetitive foot shocks were applied to demonstrators during 0–4 min. Error bars represent standard error of the mean. *Right*, the percentage of immobility time during the 4-min shock period. Box and whisker plots indicate medians (line within box), first and third quartiles (bounds of box), and the distributions of 10 and 90% (whiskers). For demonstrators, the data from the FO-only group and the PS + FO group were pooled because there was no significant intergroup difference. no-PS/FO versus FO-only: $t_{38} = 5.15$, $P < 1.0 \times 10^{-4}$; no-PS/FO versus PS-only: $t_{38} = 1.68 \times 10^{-2}$, $P = 0.497$; no-PS/FO versus PS + FO: $t_{38} = 7.44$, $P < 1.0 \times 10^{-4}$; FO-only versus PS-only: $t_{38} = 5.02$, $P < 1.0 \times 10^{-4}$; FO-only versus PS + FO: $t_{38} = 4.72$, $P < 1.0 \times 10^{-4}$; PS-only versus PS + FO: $t_{38} = 7.40$, $P < 1.0 \times 10^{-4}$; **$P < 0.01$, $t$-based bootstrap test after Kruskal–Wallis test, $n = 20$ mice per group. **c** Effects of intraperitoneal (i.p.) injection of ethanol and the oxytocin receptor antagonist L-368,899 on observational immobility. Saline versus L-368,899: $t_{42} = 5.02$, $P < 1.0 \times 10^{-4}$; saline versus EtOH: $t_{42} = 4.88$, $P < 1.0 \times 10^{-4}$; saline versus EtOH + L-368,899: $t_{42} = 3.87$, $P < 1.0 \times 10^{-4}$; L-368,899 versus EtOH: $t_{42} = 12.1$, $P < 1.0 \times 10^{-4}$; L-368,899 versus EtOH + L-368,899: $t_{42} = 9.08$, $P < 1.0 \times 10^{-4}$; EtOH versus EtOH + L-368,899: $t_{42} = 0.251$, $P = 0.387$; **$P < 0.01$, $t$-based bootstrap test after one-way ANOVA, $n = 22$ mice per group

analyses of temporal activity by fluorescent in situ hybridization (catFISH)[31], which can discriminate neurons that are activated during two time windows, the firsthand pain (primary shock) and the vicarious pain (fear observation), by taking advantage of the intracellular localization of the immediate early gene *Arc*; in response to neuronal activity, *Arc* mRNA appears in the nucleus within 5 min and moves to the cytoplasm 20–45 min later. Therefore, in the experimental schedule depicted in Fig. 2a, nuclear and cytoplasmic *Arc*-positive (Nuc$^+$ and Cyto$^+$, respectively) cells are expected to represent neurons that were activated by the fear observation and the priming shock, respectively. We first analyzed the *Arc* expression in ACC neurons (Fig. 2b). The percentages of Nuc$^+$ cells were consistent with those reported in a previous study[32], and did not differ between the PS + FO and FO-only groups (Fig. 2c), suggesting that the facilitatory effect of the priming shock was not mediated simply by an increase in ACC neuronal activity. We then computed the overlap score for nuclear/cytoplasmic double-positive (Double$^+$) cells (Fig. 2d), which represents the extent to which neurons were commonly activated by both the priming shock and the fear observation. The PS + FO group had a significantly higher overlap score than the FO-only control group (PS + FO: $57.9 \pm 3.5$, FO-only: $16.0 \pm 1.4$, $t_{10} = 11.2$, $P = 5.70 \times 10^{-2}$, $n = 6$ mice per group). Consistent with the behavioral data, the scores of the PS + FO group were decreased and increased by intraperitoneal L-368,899 and ethanol, respectively (Fig. 2e), and L-368,899 did not affect ethanol-induced increases in scores (Fig. 2e).

We also analyzed *Arc* mRNA in the anterior insular cortex, the basal/lateral amygdala (BLA), and the dorsal hippocampal CA1 region (Supplementary Fig. 9), but unlike in the ACC, we failed to find significant overlap scores (Fig. 2f), though these brain regions exhibited significant percentages of Double$^+$ cells; note that the overlap score was defined such that it excluded the possible effects of population sizes[33]. Consistent with that result, local injections of 4.1 μg/side L-368,899 and 7.0 μg/side ethanol into the ACC were sufficient to reduce and enhance observational fear response, respectively (Fig. 3a). Moreover, individual-animal analyses demonstrated that the duration of defensive immobility during observation was positively correlated with the overlap scores of ACC neurons, but not with the percentages of Cyto$^+$ or Nuc$^+$ cells (Fig. 3b–d).

Although the overlap scores in the BLA did not differ between the PS-only and PS + FO groups (Fig. 2f), the BLA receives anatomically direct afferents from the ACC, and this neural pathway is a part of the neuronal substrate for observational fear learning[15,34]. Thus, the overlapping ACC activity may be transmitted to the BLA. We retrogradely labeled the neurons projecting to the BLA by injecting Alexa 594-conjugated cholera toxin subunit B (CTB, 0.5 μg/side) into the BLA and conducted *Arc* catFISH for ACC neurons (Fig. 4a–d). CTB-positive cells were more frequently observed in Cyto$^+$ cells than in nucleic-only *Arc*-positive cells or double-negative cells (Fig. 4e, $n = 4$ mice). Thus, ACC neurons that projected to the BLA were preferentially recruited by a priming shock, potentially indicating

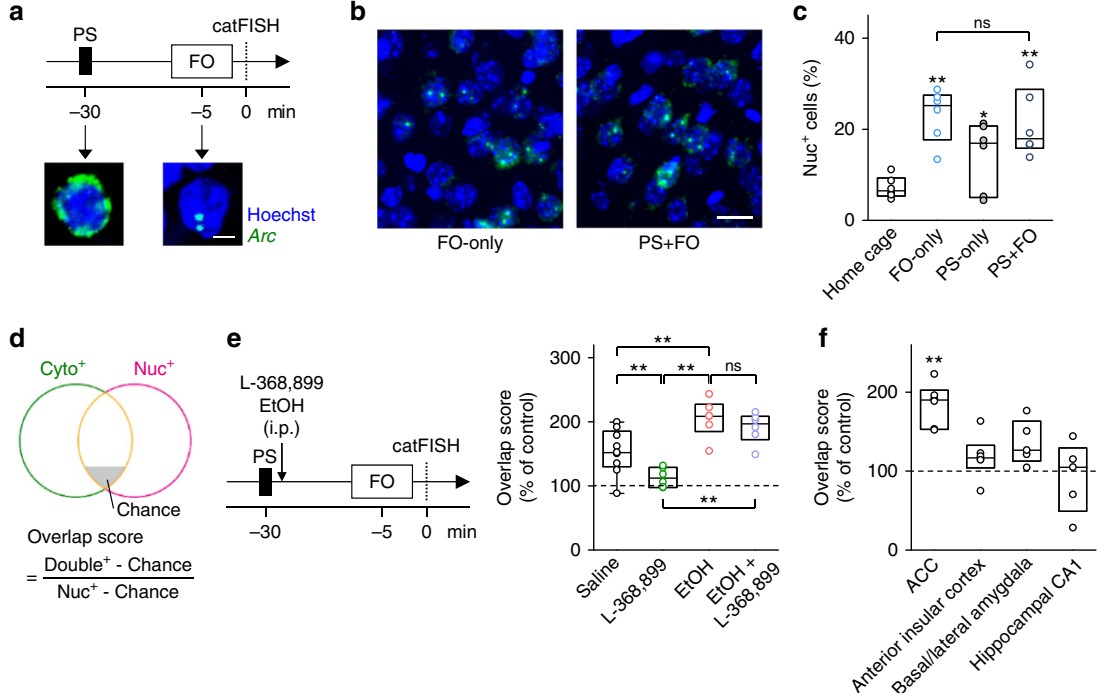

**Fig. 2** Firsthand and vicarious pain share neuronal representations in the ACC. **a** *Top*, experimental procedures for temporal activity mapping using *Arc* catFISH in the fear observation (FO) test with a single priming foot shock (PS). *Bottom*, confocal images of in situ *Arc* mRNA localization with Hoechst counterstaining. Scale bar is 5 μm. **b** Representative confocal images of *Arc* expression in ACC cells of observer mice that received (PS + FO) or did not receive (FO-only) a priming shock. Scale bar is 20 μm. **c** Percentages of cells that expressed *Arc* in the nuclei (Nuc$^+$) did not differ between the FO-only and PS + FO groups. Home cage versus FO-only: $t_{10} = 6.22$, $P < 1.0 \times 10^{-4}$; home cage versus PS-only: $t_{10} = 2.18$, $P = 1.32 \times 10^{-2}$; home cage versus PS + FO: $t_{10} = 3.87$, $P < 1.0 \times 10^{-4}$; FO-only versus PS + FO: $t_{10} = 0.477$, $P = 0.296$; *$P < 0.05$, **$P < 0.01$, t-based bootstrap test after one-way ANOVA, $n = 6$ mice per group. **d** Venn diagram representing the population overlap in ACC cells that were activated during priming shock (Cyto$^+$) and/or observation (Nuc$^+$). Chance was computed as the product of the probabilities of Cyto$^+$ and Nuc$^+$. **e** Effects of systemic injection of 5 mg/kg L-368,899 and 1.5 g/kg ethanol on the overlaps of ACC neuron populations. The overlap scores in the PS + FO group were normalized to the PS-only control group. Saline versus L-368,899: $t_{13} = 3.08$, $P = 1.80 \times 10^{-3}$; saline versus EtOH: $t_{13} = 3.07$, $P = 2.30 \times 10^{-3}$; saline versus EtOH + L-368,899: $t_{13} = 2.44$, $P = 6.42 \times 10^{-3}$; L-368,899 versus EtOH: $t_{10} = 6.76$, $P < 1.0 \times 10^{-4}$; L-368,899 versus EtOH + L-368,899: $t_{10} = 6.75$, $P = 1.01 \times 10^{-4}$; EtOH versus EtOH + L-368,899: $t_{10} = 0.944$, $P = 0.149$; **$P < 0.01$, t-based bootstrap test after one-way ANOVA, $n = 6$–9 mice per group. **f** The overlap scores in the ACC, the anterior insular cortex, the basal/lateral amygdala (BLA), and the dorsal hippocampal CA1 region in the PS + FO group were normalized to those of the PS-only group. **$P < 0.01$ versus the PS-only group, $n = 5$–6 mice per group

that the increased overlap of ACC neuron ensembles reflects increased functional connectivity between the ACC and BLA.

**Ethanol shifts the E/I-balance in ACC neurons**. To examine how ethanol modulates ACC neuronal activity, we recorded miniature excitatory and inhibitory postsynaptic currents (mEPSCs and mIPSCs, respectively) from layer II/III ACC pyramidal neurons in acute brain slices and bath-applied 50 mM ethanol (Fig. 5a), which corresponds to the normal intrabrain concentrations after systemic injection of ethanol[35]. Ethanol decreased the amplitudes and frequencies of mEPSCs and increased the frequencies of mIPSCs, whereas it did not alter the amplitudes of mIPSCs (Supplementary Fig. 10). Overall, ethanol decreased the mEPSC conductances (Fig. 5b, left) and increased the mIPSC conductances (Fig. 5b, middle), thereby reducing the net excitatory to inhibitory (E/I) ratio and tipping the balance toward inhibition (Fig. 5b, right). To examine how the E/I-balance shift affects socially evoked fear memory recall, we injected 79 ng/side clonazepam, a traditional benzodiazepine, or 0.3 μg/side picrotoxin, a γ-aminobutyric acid (GABA)$_A$-receptor inhibitor, into the ACC. Note that clonazepam is reported to reduce the synaptic E/I ratios and thereby ameliorate social behavioral deficits[36]. As we expected, clonazepam increased observational fear response in control mice, whereas picrotoxin reduced it

(Fig. 5c). Picrotoxin also abolished ethanol-enhanced observational fear response (Fig. 5c).

**Ethanol rescues social impairment of poly(I:C) mice**. Finally, we examined observational fear in a mouse model of maternal immune activation[37]. In humans, maternal immune activation is a known risk factor for autism spectrum disorder. We treated pregnant mice with poly-inosine:cytosine (poly(I:C)), a synthetic analogue of double-stranded RNA that mimics a molecular pattern of viral infection, and tested the offspring in the fear observation test. The priming shock failed to enhance observational fear response in these poly(I:C) mice (Fig. 6a). However, intraperitoneal ethanol administration in poly(I:C) mice restored observational fear response to levels comparable to those in control mice (Fig. 6b). Poly(I:C) mice exhibited normal defensive immobility during their own contextual fear conditioning and higher defensive immobility during subsequent memory tests (Supplementary Fig. 11). Thus, neither fear expression nor fear learning ability was impaired in poly(I:C) mice. *Arc* catFISH revealed that poly(I:C) mice exhibited a lower ACC overlap score than control mice (Fig. 6c), whereas there were no effects on the percentages of either Cyto$^+$ or Nuc$^+$ cells per se (Supplementary Fig. 12). The overlap score of poly(I:C) mice was increased by intraperitoneal ethanol (Fig. 6d).

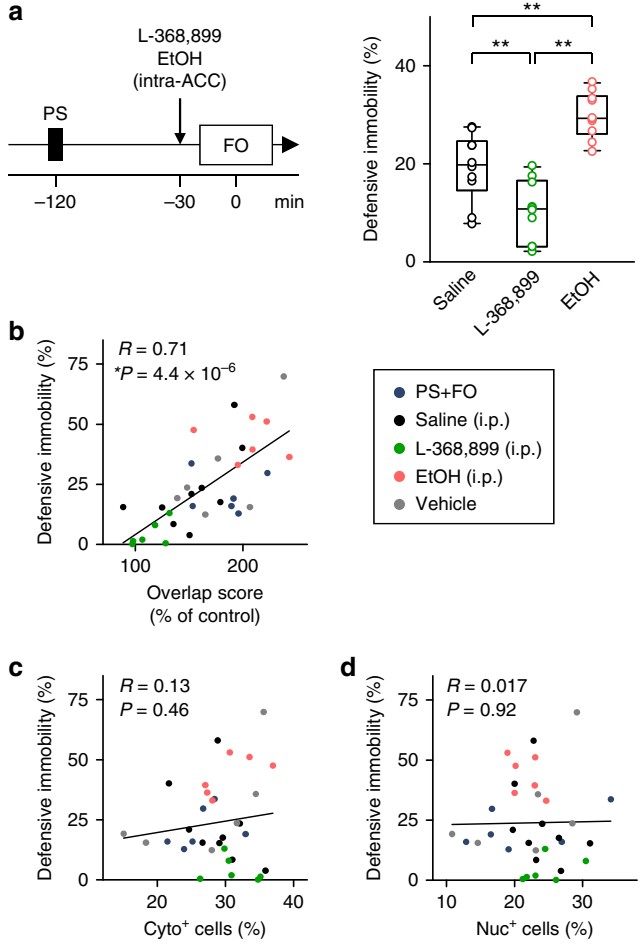

**Fig. 3** The overlap scores in the ACC are correlated with defensive immobility. **a** Effects of bilateral intra-ACC microinjection of 4.1 μg/side L-368,899 and 7.0 μg/side ethanol on observational immobility. Saline versus L-368,899: $t_{18} = 3.03$, $P = 9.03 \times 10^{-4}$; saline versus EtOH: $t_{18} = 4.07$, $P < 1.0 \times 10^{-4}$; L-368,899 versus EtOH: $t_{18} = 7.98$, $P < 1.0 \times 10^{-4}$; **$P < 0.01$, $t$-based bootstrap test after one-way ANOVA, $n = 10$ mice per group. **b–d** Correlation plots of defensive immobility during observation against the overlap scores (**b**) and the percentages of Cyto+ cells (**c**) and Nuc+ cells (**d**) suggest that observational fear is shaped by a commonly shared representation rather than the activity level per se of ACC neurons. These data include mice in the PS + FO group in Fig. 2c, the drug-treated groups in Fig. 2e, and the vehicle group in Fig. 6c. Each dot indicates a single mouse, and the lines are the best fits using the least-squares method. $n = 33$ mice

## Discussion

In the present study, we discovered that neuronal representations of experienced and observed pain overlapped at the single-cell level and were bidirectionally manipulable by administration of ethanol or an oxytocin receptor antagonist, which induced a corresponding change in defensive immobility during fear observation.

Recent studies have demonstrated overlapping neuronal ensembles across associative memories in various learning paradigms[38,39], and the overlapping representations are indispensable for a temporal link between discrete memories[39]. Our study extends this notion to an inter-individual link of experiences. We observed that the degrees of defensive immobility during observation were positively correlated with the overlap scores of ACC neurons but not with the absolute numbers of activated neurons,

consistent with a report showing that the strength of fear memory is not coded by the overall size of memory trace in the lateral amygdala[40]. Thus, overlapping populations of ACC neurons may integrate firsthand and vicarious pain information and facilitate socially evoked fear memory recall.

Our findings suggest that overlapping neuronal representations are shaped by neuronal E/I balance, consistent with other studies about learning and memory. The E/I balance in individual neurons is a fundamental determinant of memory allocation[41,42]. The E/I balance regulates the neuronal overlap and thereby determines whether two memories are integrated or segregated[43]. A computational study has also shown that balanced E/I ratios are required for noise-robust neuronal selection[44]. Moreover, imbalanced E/I ratios are one of the major biological characteristics of human autism spectrum disorders, in which the number of parvalbumin-expressing interneurons and the GABA concentration in the cerebral cortex are reduced[45,46]. GABAergic signaling is also disrupted in mouse models of autism[47,48]. In normal mice, an artificial elevation of the E/I balance leads to social deficits[36,49,50]. However, the mechanisms linking the E/I imbalance and social impairments remain unclear. Based on our results, we suggest that ethanol reduced the E/I ratio, increased the shared neuronal representations for experienced and observed pain, and enhanced observational fear in both normal and autism model mice, thus possibly providing a cell-level framework for both sociality and social impairments in developmental disorders. However, our data must be interpreted with caution. We performed only pharmacological manipulations on the excitability of neuronal circuits of the ACC and did not provided a direct link between the E/I balance and social behavior. Moreover, we found that injection of picrotoxin alone abolished the observational fear response. Therefore, it is possible that picrotoxin caused dysfunction of the ACC neuronal circuits, rather than a modulation of the E/I balance, and impaired social behavior.

We rule out the possibility that the facilitatory effect of ethanol on defensive immobility during fear observation resulted merely from ethanol-induced sedation because ethanol-increased locomotor activity and had no effect on immobility time in the open field test at the dose (1.5 g/kg) used in our study. Consistent with our observations, ethanol is known to increase locomotor frequency but does not alter immobility in the open field test at similar doses[30]. Another concern of using ethanol was the acute effect on memory. In humans, ethanol administered after training is reported to facilitate the ability to learn various tasks[51–53]; the results from animal studies are contradictory. Posttraining ethanol can promote learning[54,55] or impair learning[56]. Thus, in our study, it is possible that ethanol-enhanced consolidation of the priming shock memory facilitated observational fear. However, we demonstrated that injection of ethanol prior to the fear observation, but not after the priming shock, promoted observational fear, suggesting that the enhanced memory consolidation cannot account for the facilitatory effect of ethanol. The defensive immobility in the PS + FO group contains an aspect of memory recall. Therefore, ethanol may increase defensive immobility by enhancing the recall of observer's own fear memory. We confirmed that ethanol did not promote fear memory recall in a classical fear-conditioning paradigm. Previous reports have also demonstrated that ethanol impairs, rather than facilitates, memory retrieval in the Morris water maze task[57] and the passive avoidance task[58]. Therefore, we believe that ethanol-increased defensive immobility was unlikely to have been produced simply by ethanol-enhanced associative memory recall. Moreover, we confirmed that a priming shock per se did not enhance the fear response and did not induce contextual associative fear memory. These results suggest that the immobility behavior in the PS + FO group is essentially different from fear memory recall in a classical

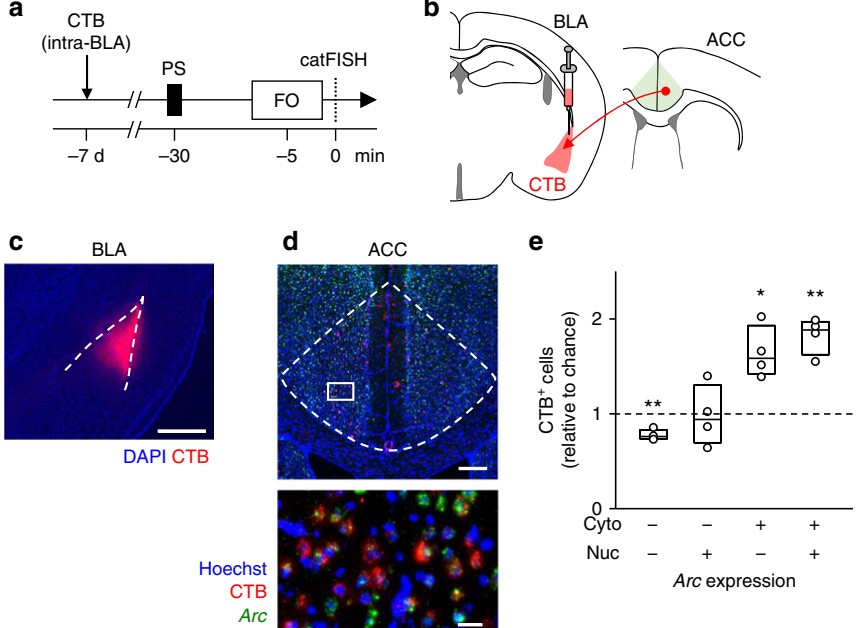

**Fig. 4** A priming shock preferentially activates ACC neurons that project to the BLA. **a, b** Experimental paradigm (**a**) and schematic illustration (**b**) of retrograde cholera toxin subunit B (CTB) labeling and *Arc* catFISH mapping. **c** A representative confocal image for CTB (red) signals in a DAPI-counterstained BLA section. The dashed area indicates the BLA. Scale bar is 500 μm. **d** Representative images for CTB (red) and *Arc* mRNA (green) signals in a Hoechst-counterstained ACC section. The boxed region in the ACC is magnified in the bottom image. The dashed area indicates the ACC. Scale bars in the top and the bottom images are 200 and 20 μm, respectively. **e** The numbers of CTB+ cells in a given *Arc* expression pattern (Cyto+, Nuc+, or Cyto +&Nuc+) are normalized to the chance level, which is defined as the product of the number of cells involved in the *Arc* pattern and the CTB-positive ratio in all Hoechst-positive ACC cells. $^*P < 0.05$, $^{**}P < 0.01$ versus the chance level, $n = 4$ mice

fear conditioning. It is intriguing to find that ethanol did not increase the defensive immobility in the FO-only group. Thus, ethanol is unlikely to enhance empathetic fear responses. A plausible interpretation is that ethanol selectively enhances a socially evoked component of the fear memory recall.

Interestingly, alcohol-dependent patients are known to exhibit a lower level of empathy[59]. This fact is apparently contradictory to our findings, but chronic ethanol treatment induces neurochemical changes in both excitatory and inhibitory synaptic transmissions in the cerebral cortex, and those changes are different from the acute effect of ethanol. Chronic ethanol increases the mRNA level of NMDA receptors[60] and decreases the mRNA level of GABA_A receptors[61], potentially shifting the E/I balance toward excitation. In fact, cortical inhibition is reduced in alcoholic patients[62]. By contrast, we found that the acute application of ethanol shifted the E/I balance toward inhibition. Therefore, ethanol seems to exert opposite effects in acute or chronic treatment.

There are also conflicting reports regarding the effects of ethanol on the perception of one's own pain. Several studies have demonstrated that systemic injection of ethanol induces analgesia that is evaluated by tail-deflection or tail-flick assays[63,64], whereas other reports have shown that similar doses of ethanol have no effect on motor responses to various intensities of electrical shocks[29] and even that ethanol induces hyperalgesia[65]. This discrepancy may be due to differences in the types or intensities of noxious stimuli delivered to animals[64]. In our study, intraperitoneal injection of 1.5 g/kg ethanol did not alter the minimal electrical shock intensity needed to induce pain-related responses or the frequency of acetic acid-induced writhing responses, suggesting that ethanol did not apparently affect pain sensitivity. Under our experimental conditions, therefore, ethanol-enhanced observational fear is unlikely attributable to a sensitized pain system.

In contrast to our findings, some studies have demonstrated that ethanol enhances aggression in mice[66,67]. These studies assessed aggressive behaviors using resident–intruder tests, in which a resident mouse attacks an unfamiliar mouse that intrudes the territory[68]. This paradigm differs from the situation in our fear observation test in which an observer and a demonstrator are cagemates and do not exhibit offensive behaviors against one another. In general, familiarity between two animals is crucial for establishing empathy-related behaviors, such as fear transmission[15], prosocial behavior[69], and social modulation of pain[12]. In humans, oxytocin augments not only in-group favoritism but also out-group derogation[25]. Ethanol may work in a similar way and facilitate socially evoked fear memory recall only to in-group members.

Also in contrast to our study showing that oxytocin increases observational fear expression, several reports have demonstrated that oxytocin attenuates fear per se through modulating the activity of the central amygdala[70,71]. These studies showed that artificial augmentation of oxytocin signaling, such as microinjection of oxytocin receptor agonist and optogenetically evoked axonal oxytocin release, in the central amygdala causes fear reduction, but they did not mention whether naturally released endogenous oxytocin affects defensive immobility. Indeed, systemic administration of L-368,899 (5 or 10 mg/kg) did not alter defensive immobility in Pavlovian conditioning[26]. A metaanalysis of human studies also found that intranasal oxytocin did not significantly influence the expression of negative emotions, including fear[72]. Therefore, it is unclear whether naturally released oxytocin in the amygdala is truly involved in fear, even though artificially elevated oxytocin signaling can attenuate fear. On the other hand, we showed that intra-ACC injection of L-368,899 is sufficient to reduce observational fear response, suggesting that oxytocin differently affects Pavlovian-conditioned fear and socially evoked fear. This difference may simply reflect

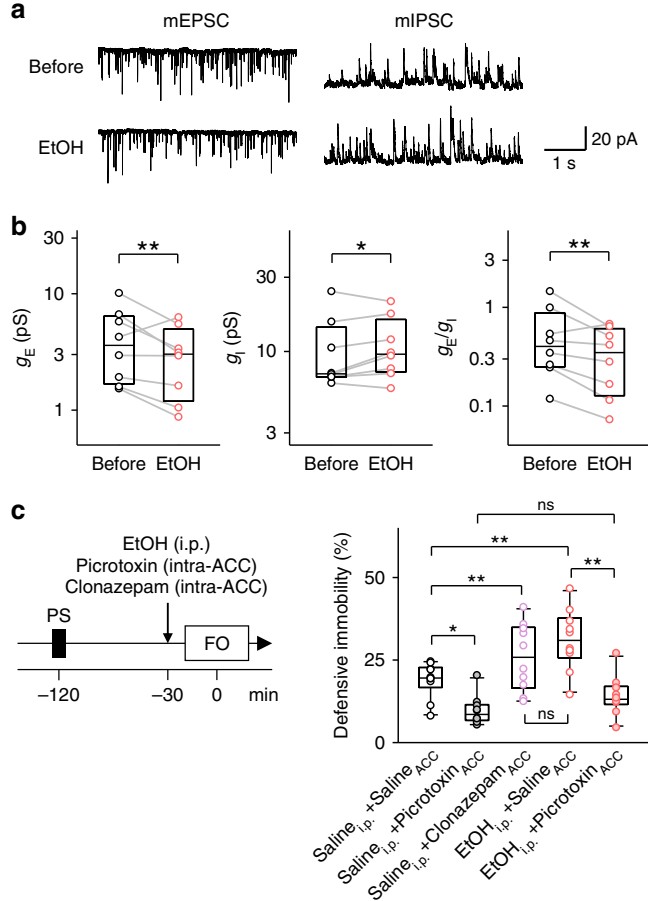

**Fig. 5** Ethanol-enhanced socially evoked fear memory recall requires E/I-balanced ACC activity. **a** Representative whole-cell traces of mEPSCs and mIPSCs in ACC layer II/III pyramidal cells in acute brain slices. **b** Ethanol decreased the mean excitatory synaptic conductance ($g_E$, *left*; $t_7 = 2.48$, $^{**}P = 4.67 \times 10^{-3}$, $t$-based bootstrap test, $n = 8$ cells), whereas it increased the inhibitory conductance ($g_I$, *middle*; $t_7 = 1.75$, $^{*}P = 2.12 \times 10^{-2}$). As a result, ethanol shifted the E/I balance toward inhibition ($g_E/g_I$, *right*; $t_7 = 3.21$, $^{**}P = 1.13 \times 10^{-3}$). **c** The facilitatory effect of intraperitoneal ethanol on socially evoked fear memory recall was blocked by intra-ACC microinjection of 0.3 μg/side picrotoxin, a GABA$_A$ receptor antagonist. Saline$_{ip}$ + saline$_{ACC}$ versus saline$_{ip}$ + picrotoxin$_{ACC}$: $t_{18} = 1.92$, $P = 2.21 \times 10^{-2}$; saline$_{ip}$ + saline$_{ACC}$ versus saline$_{ip}$ + clonazepam$_{ACC}$: $t_{18} = 4.23$, $P < 1.0 \times 10^{-4}$; saline$_{ip}$ + saline$_{ACC}$ versus EtOH$_{ip}$ + saline$_{ACC}$: $t_{18} = 3.58$, $P = 3.60 \times 10^{-4}$; saline$_{ip}$ + picrotoxin$_{ACC}$ versus EtOH$_{ip}$ + picrotoxin$_{ACC}$: $t_{18} = 1.43$, $P = 6.36 \times 10^{-2}$; saline$_{ip}$ + clonazepam$_{ACC}$ versus EtOH$_{ip}$ + saline$_{ACC}$: $t_{18} = 1.18$, $P = 0.100$; EtOH$_{ip}$ + saline$_{ACC}$ versus EtOH$_{ip}$ + picrotoxin$_{ACC}$: $t_{18} = 4.87$, $P < 1.0 \times 10^{-4}$; $^{*}P < 0.05$, $^{**}P < 0.01$, $t$-based bootstrap test after Kruskal–Wallis test, $n = 10$ mice per group

the difference between the brain regions, and it is also possible that oxytocin-reduced fear is surpassed by oxytocin-induced facilitation of socially evoked fear.

In summary, we demonstrated that ACC neuronal representation of self-experienced pain and vicarious pain overlaps at the single-cell level and that this overlapping representation may underlie shared affective states between individuals. Moreover, our study revealed that the overlapping representation was increased by acute administration of ethanol. Our data were obtained exclusively from a fear observational system in mice, and extrapolation to other empathetic states requires care; however, our findings provide insights into the role of ethanol in sociality.

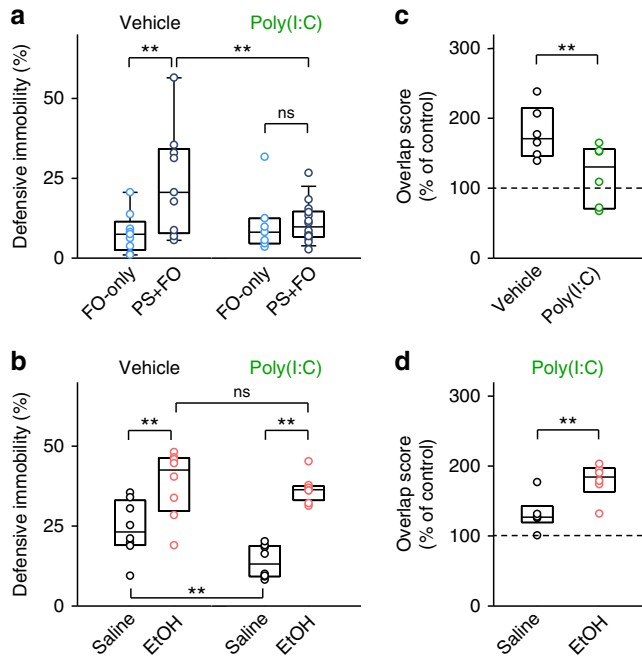

**Fig. 6** Ethanol rescues impairment of socially evoked fear memory recall in poly(I:C) mice. **a** A priming foot shock did not increase the percentage of defensive immobility time in poly(I:C) mice. FO-only versus PS + FO in vehicle: $t_{16} = 2.70$, $P = 2.21 \times 10^{-3}$; FO-only versus PS + FO in poly(I:C): $t_{19} = 5.16 \times 10^{-2}$, $P = 0.456$; vehicle versus poly(I:C) with PS + FO: $t_{21} = 2.22$, $P = 3.83 \times 10^{-3}$; $^{**}P < 0.01$, $t$-based bootstrap test after one-way ANOVA, $n = 7$–14 mice per group. **b** Intraperitoneal ethanol injection enhanced socially evoked fear memory recall in poly(I:C) mice. Saline versus EtOH in vehicle: $t_{14} = 2.92$, $P = 2.24 \times 10^{-3}$; saline versus EtOH in poly(I:C): $t_{14} = 9.69$, $P < 1.0 \times 10^{-4}$; vehicle versus poly(I:C) with saline: $t_{14} = 2.91$, $P = 2.69 \times 10^{-3}$; vehicle versus poly(I:C) with EtOH: $t_{14} = 0.459$, $P = 0.289$; $^{**}P < 0.01$, $t$-based bootstrap test after one-way ANOVA, $n = 8$ mice per group. **c** The overlap score of anterior cingulate cortex (ACC) neuronal populations in poly(I:C) mice was lower than that in the vehicle-treated, typically developing group. $t_{10} = 2.54$, $^{**}P = 1.07 \times 10^{-3}$; $t$-based bootstrap test, $n = 6$ mice per group. **d** Systemic ethanol increased the overlap score in the ACC in poly(I:C) mice. $t_{10} = 3.28$, $^{**}P = 2.18 \times 10^{-4}$; $t$-based bootstrap test. $n = 6$ mice per group

## Methods

**Animals**. Male 4- to 12-week-old C57BL/6J mice (SLC, Shizuoka, Japan) were housed under conditions of controlled temperature and humidity (23 ± 1 °C, 55 ± 5%), maintained on a 12:12-h light/dark cycle with the lights on from 7:00 a.m. to 7:00 p.m., and had access to food and water ad libitum. Animal experiments were performed with the approval of the animal experiment ethics committee at the University of Tokyo (approval numbers: P24-70 and P29-11) and according to the University of Tokyo guidelines for the care and use of laboratory animals. These experimental protocols were carried out in accordance with the Fundamental Guidelines for Proper Conduct of Animal Experiment and Related Activities in Academic Research Institutions (Ministry of Education, Culture, Sports, Science and Technology, Notice No. 71 of 2006), the Standards for Breeding and Housing of and Pain Alleviation for Experimental Animals (Ministry of the Environment, Notice No. 88 of 2006) and the Guidelines on the Method of Animal Disposal (Prime Minister's Office, Notice No. 40 of 1995).

**Drugs**. The doses of ethanol (1.5 g/kg for systemic injection, 7.0 μg/side for intra-ACC injection, Wako) were determined based on the blood ethanol concentrations observed with daily alcohol consumption in humans[27,28], and on previous studies of rodent behavior[55,73]. The concentration of ethanol (50 mM) in in vitro experiments using brain slices was based on the normal intrabrain concentration after systemic injection of ethanol[35]. The doses of other drugs were also chosen with reference to previous reports of rodent behavior (MK801: 0.1 mg/kg[74], Tocris Bioscience; L-368,899: 5 mg/kg, 4.1 μg/side[75], Tocris Bioscience; picrotoxin: 0.3 μg/side[76], Nacarai Tesque; clonazepam: 79 ng/side[77], Wako). For behavioral experiments, all drugs except for clonazepam were diluted in 0.9% saline. Clonazepam

was diluted in 0.9% saline with 0.1% DMSO (Nacarai Tesque) because it is hydrophobic. The injected volumes were 10 ml/kg for systemic injection and 0.5 µl/side for intra-ACC injection.

**Fear observation**. For the fear observation test, two cagemates were defined as either the observer or demonstrator and were individually placed in chambers that were partitioned by a transparent plastic plate. The chamber for the observer had plastic walls and a flat white floor (19 cm in width, 16 cm in depth, and 27 cm in height). The chamber for the demonstrator (shock chamber) had plastic walls and a metal grid floor (18 cm in width, 11 cm in depth, and 11 cm in height) connected to a shock scrambler (SGA-2010, O'HARA, Tokyo, Japan). One mouse was placed in each chamber for 5 min, and then the demonstrator received a 1-s foot shock (1 mA) every 12 s for 4 min. During the fear observation period, the animals were recorded at 2 Hz using a top-view digital camera. Immobility was automatically identified using a custom-made MATLAB (R2013b; MathWorks, Natick, Massachusetts, USA) routine[74]. After denoising the image, the mouse in each video frame was binarized, and the body motion was detected by calculating the number of pixels in which the binary values changed between two successive video frames. Immobility time was defined based on the total number of frames in which the number of pixel changes was below the threshold. The threshold was determined such that the detected immobility was comparable to that obtained manually by three well-trained experimenters. In the PS + FO group, observers received a single priming foot shock (1 mA, 2 s) prior to fear observation. A priming shock was given immediately after the observer was placed in a shock chamber. Immediately after the shock delivery, the observer was returned to its home cage.

**Tail pinch**. The base of the mouse's tail was pinched for 5 s using an artery clip (4.5 cm in length). The latency until the mice tried to remove the tail clip was recorded, and only those that attempted to remove the clip within 1 s were used in the fear observation test.

**Forced swimming**. Individual mice were forced to swim inside a vertical Plexiglas cylinder (inner $\phi = 12$ cm). The water temperature was $22 \pm 1$ °C, the depth was 15 cm, and the above-water wall height was 8 cm. The mice were kept in the water for 15 min, and within that time period, all mice began spending 60% of their time immobile. The mice were then returned to their home cages. The water was changed for each mouse.

**Open field test**. In each experiment, a mouse was placed in the center of a square, white plastic box (40 cm in width, 40 cm in depth, and 30 cm in height) with an open top. The test was conducted under the room brightness of 165 lux. A camera was installed above the center of the field to monitor the instantaneous position of the mouse. The total distance traveled for 10 min by each mouse was measured. Ethanol (1.5 g/kg) or saline was intraperitoneally injected 30 min before the test.

**Elevated plus-maze test**. Individual mice were placed in the center of a maze with four arms arranged in the shape of a plus sign. The maze consisted of a central quadrangle (8 cm in width and 8 cm in length), two opposing open arms (25 cm in length and 8 cm in width), and two opposing closed arms. These four arms were identical in size, but the closed arms were equipped with 25-cm high walls at both the sides and the far end. The floorboard was made of white plastic, and elevated 25 cm above the ground, and the walls were made of opaque gray plastic. At the beginning of each trial, the mice were placed on the central quadrangle facing an open arm. The apparatus was illuminated by ceiling lights at 165 lux. The movements of the mice during a period of 5 min were recorded by a camera positioned above the center of the maze. The numbers of entries into the open arms and the closed arms and the time spent in each arm were manually determined. An entry into an arm was defined as placement of all four paws on that arm. The number of open-arm entries is expressed as a percentage of the total arm entries, i.e., open-arm entries/(open-arm entries + closed-arm entries) × 100.

**Shock threshold test for pain sensitivity**. In each experiment, a mouse was placed in a shock chamber consisting of a plastic box (18 cm in width, 11 cm in depth, and 11 cm in height), four transparent walls and a metal grid floor connected to a shock scrambler (SGA-2010, O'HARA, Tokyo, Japan). Repetitive 1-s foot shocks were manually applied to the mouse at intervals of at least 30 s, and a well-trained experimenter observed the behavioral responses. Shock intensities started at 10 µA and increased at every 10-µA step until both a detectable reflex muscle response and vocalization were induced. The minimal intensity of shock necessary for inducing these responses was defined as the shock threshold. Ethanol (1.5 g/kg) or saline was intraperitoneally injected 30 min before the test.

**Acetic acid-induced writhing test**. Pain sensitivity was evaluated by measuring the acetic acid-induced writhing responses[12,78]. Acetic acid (0.9%) was intraperitoneally injected at a volume of 10 ml/kg into mice that were placed in a plastic cage (18 cm in width, 11 cm in depth, and 11 cm in height) 5 min before the test. Mice were habituated to these cages for 30 min before injection, and their movements were video-recorded for 10 min. A stretching behavior of the hind limbs

accompanied by a contraction of the abdominal muscles was defined as a writhing response. From each video, a total of 30 periods (each 5 s in length) were extracted at an interval of 20 s. A blinded observer judged whether the writhing behaviors were present or absent in these 5-s video clips. The writhing frequency was expressed as the percentage of video clips with writhing responses.

**Sample preparation and fluorescence in situ hybridization**. Mice were sacrificed 5 min after the fear observation test was finished, and their brains were removed and frozen quickly. In situ hybridization was performed as followings[33]. Coronal brain sections (20 µm) were hybridized with digoxigenin-labeled riboprobe (1.5 µg/ml). The signals were detected with an anti-digoxigenin-HRP antibody (1:500; Roche, Cat# 11207933910, RRID: AB_514500) and then with biotinyl tyramide (1:5000) with Alexa 488-conjugated or Alexa 594-conjugated streptavidin (Invitrogen). The nuclei were counterstained with Hoechst. Z-stacks of 1-µm-thick optical sections were acquired with a CV1000 confocal microscope using a 40× objective lens (Yokogawa). The majority of cells exhibited whole, large nuclei stained diffusely with the Hoechst dye. Only these putative neurons were included in the analysis. The designation "nuclear positive" or "Nuc$^+$" indicates neurons that exhibited one or two of the intensely fluorescent intranuclear bodies. The designation "cytoplasmic positive" or "Cyto$^+$" indicates neurons that contained perinuclear/cytoplasmic labeling across multiple optical sections. An overlap score between neuronal populations that were active during the priming shock and the fear observation was defined as $(D -$ chance level$)/(N -$ chance level$)$, in which $C$ (percentage of total Cyto$^+$ cells) represents cytoplasm only (%) plus Nuc$^+$&Cyto$^+$ (%); $N$ (percentage of total Nuc$^+$ cells) represents nucleus only (%) plus Nuc$^+$&Cyto$^+$ (%); $D$ represents percentage of cells in which *Arc* signals were observed in both the cytoplasm and nucleus; the chance level was defined as $C \times N$ /100.

**Surgery for microinjection**. Under intraperitoneal xylazine (10 mg/kg) and pentobarbital (25 mg/kg) anesthesia, mice were bilaterally implanted with 26-gauge stainless steel guide cannulae (Plastics One, Roanoke, VA, United States) into the ACC (A/P 1.0 mm, L/M ± 0.2 mm, D/V 2.0 mm) or BLA (A/P 1.3 mm, L/M ± 3.3 mm, D/V 4.8 mm). These cannulae were fixed to the skull using a mixture of acrylic and dental cement, and 33-gauge dummy cannulae were then inserted into the guide cannulae to prevent clogging. Mice were given at least seven days of postoperative recovery time. For microinjection, an infusion cannula was carefully inserted into each guide cannula. Drug solution or an equivalent volume of saline was bilaterally infused at a rate of 0.25 µl/min under free-moving conditions in a plastic cage (18 cm in width, 11 cm in depth, and 11 cm in height). The infusion cannulae were left in place for 2 min to avoid backward diffusion of the solution into the guide cannulae.

**In vitro electrophysiology**. Acute neocortical slices containing the ACC area were prepared from postnatal week 3–4 mice. Mice were anesthetized with isoflurane and decapitated, and the brain was coronally sliced (400 µm thick) in an ice-cold oxygenated cutting solution consisting of (in mM) 222.1 sucrose, 27 NaHCO$_3$, 1.4 NaH$_2$PO$_4$, 2.5 KCl, 1 CaCl$_2$, 7 MgSO$_4$, and 0.5 ascorbic acid using a vibratome. Slices were allowed to recover for at least 1 h while submerged in a chamber filled with oxygenated artificial cerebrospinal fluid (aCSF) at room temperature. aCSF consisted of (in mM) 27 NaCl, 26 NaHCO$_3$, 1.6 KCl, 1.24 KH$_2$PO$_4$, 1.3 MgSO$_4$, 2.4 CaCl$_2$, and 10 glucose. Recordings were performed in a submerged chamber perfused at 3 ml/min with oxygenated aCSF at 34–37 °C. Whole-cell patch-clamp current-clamped recordings were obtained, before and during bath application of 50 mM ethanol or 500 µM 6,7-dinitroquinoxaline-2,3-dione (DNQX), from layer II/III ACC pyramidal cells, which were visually identified under an infrared differential interference contrast microscope (DAGE-MTI IR-1000). Patch pipettes (3–6 MΩ) were filled with a Cs-based solution consisting of (in mM) 130 CsMeSO$_4$, 10 CsCl, 10 HEPES, 10 phosphocreatine, 4 MgATP, 0.3 NaGTP, and 10 QX-314. Fast-spiking interneurons were rejected based on their short time constants (≤2.0 ms) and nonadaptive spiking patterns. The synaptic conductance was calculated by dividing the charge (the time integral of excitatory/inhibitory postsynaptic currents) by the driving force (clamped voltage)[42]. This measure is believed to reflect general synaptic efficacy, including the amplitudes, durations, and frequencies of synaptic events[79].

**Poly(I:C) preparation and gestational exposure**. A maternal immune activation mouse model was established according to Naviaux et al.[80]. Briefly, pregnant dams received two doses of intraperitoneal poly(I:C) (potassium salt; Sigma-Aldrich, St. Louis, MO, USA, Cat# P9582) by injection (0.25 U/g [3 mg/kg] on E12.5 and 0.125 U/g [1.5 mg/kg] on E17.5), and the offspring were used for the experiments. As a control group, saline was injected into pregnant females (referred to as the Vehicle group). In the fear observation test, naïve mice of the same age were used as demonstrators. Each demonstrator was pair-housed with an observer mouse for 1 week before the test.

**Contextual fear conditioning**. Fear conditioning was performed in a shock chamber consisting of a plastic box (18 cm in width, 11 cm in depth, and 11 cm in height) with transparent walls and a metal grid floor connected to a shock scrambler (SGA-2010, O'HARA, Tokyo, Japan). Mice received a foot shock

(1 mA, 1 s) 300 s after placement in the chamber and received three additional shocks of the same intensity every 60 s. They were returned to the home cage 60 s after the last shock. The animals were replaced in the same chamber 2 and 24 h later, and their movements were video-recorded for 5 min. Immobility time was calculated as described above.

**Data analysis**. The Shapiro–Wilk test was used to confirm the normality of data distributions. The Student $t$ test, paired $t$ test, Mann–Whitney $U$ test, and $t$-based bootstrap test were used for comparisons between two groups. One-way analysis of variance, Kruskal–Wallis test and $t$-based bootstrap test were used for comparisons among more than two independent groups. Statistical significance was set at $P < 0.05$.

## Data availability

The data that support the findings of this study are available from the corresponding author upon reasonable request.

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

## Acknowledgments

We thank Dr. Manabu Makinodan for providing us with the earliest idea of this research while drinking alcohol together. This work was supported by Grants-in-Aid for Scientific Research (18H05525) and by the Human Frontier Science Program (RGP0019/2016). This work was conducted partly as a program at the International Research Center for Neurointelligence (WPI-IRCN) of The University of Tokyo Institutes for Advanced Study at The University of Tokyo.

## Author contributions

T.S. and Y.I. designed the study and wrote the manuscript. T.S. conducted the behavioral experiments and data analyses. S.I. conducted the immunohistochemistry. M.O. and K.O. conducted the electrophysiological experiments. K.O. prepared the MIA mouse model. All authors discussed the results and commented on the manuscript.

## Additional information

**Competing interests:** The authors declare no competing interests.

