## [Peer Review File · Nature Communications]

Reviewers' comments:

Reviewer #1 (Remarks to the Author):

In recent years, the research into the biological basis of empathy has been expanded to animal models, and observational fear has been recognized as a useful model for assessing affective empathy in rodents. Observational fear is defined as social transfer of fear from one individual to another with social cues in the absence of direct aversive stimuli. Thus, mice can be conditioned for fear without receiving direct foot shocks.

In this manuscript, Sakaguchi et al., evaluated the contribution of alcohol to vicarious freezing in observational fear behavior that was primed by a prior foot shock experience. The authors found that ethanol strengthened observational fear response which was primed by a prior shock experience, whereas it did not affect the unprimed spontaneous observational fear response. An oxytocin receptor antagonist did not significantly affect the ethanol-enhanced freezing in the primed observational fear, whereas it reduced the freezing in the primed observational fear in the absence of ethanol. Next, they employed the Arc mRNA activity mapping, and identified that some ACC neurons were commonly activated by both the priming shock (Priming) and the fear observation (FO), and the degree of this overlap was positively correlated with the degree of freezing. In slice experiments on ACC pyramidal neurons, they further demonstrated that ethanol shifted the excitation/inhibition balance toward to inhibition. Experiments by utilizing Arc mRNA mapping combined with retrograde labelling showed that the ACC \rightarrow BLA projecting neurons were preferentially those activated by a priming shock (with/without overlap with fear observation) but not by fear observation alone. They also showed that ethanol injection increased observational fear in the poly(I:C)-induced autism mouse model. The authors concluded that alcohol enhances social transfer of fear in association with reduced E/I ratio in the ACC. Overall this work is interesting. However, there are major concerns in the design of the behavioral task, which limit the interpretation of the findings with regard to empathy.

1. In the behavioral paradigm using observer mice with a prior shock experience, the authors found that a single priming shock (Priming) enhanced freezing of the observer mouse upon observation of a demonstrator receiving similar foot shocks (FO), and ethanol could further enhance this facilitating effect of the priming shock. Thus, the authors conclude that ethanol selectively strengthens the social transmission of fear. However, in their experimental protocol it should be noted that the observer mice experience two different sources of distress that may induce fear behavior: one through observation of demonstrator's distress without knowing the origin of the distress, and the other through remembering its own foot-shock experience on the grid (i.e., observation-induced recall of its own fear experience). While the former may involve social transmission of the affective state of the demonstrator mouse (empathy), the latter may primarily involve fear memory recall. Therefore, the results of ethanol experiments may mean that only the fear memory recall (but not empathy) is enhanced by ethanol (also stated below in 3). This perspective will inevitably change the scope of the work fundamentally.

2. In Fig 1c, an oxytocin antagonist suppressed the fear response of the Priming+FO mice. Indeed, comparing the results between Fig1b and 1c, it is noted that the level of the remaining fear response after suppression by an oxytocin antagonist in the Priming+FO group seems similar to the difference in fear level between the Priming+FO group and the FO only group, revealing the sensitivity of the FO only group to an oxytocin antagonist. This result is consistent with the known role of oxytocin in empathy, and may exclude the priming-enhanced fear as a part of empathy behavior. The authors may want to confirm the effect of oxytocin antagonists on the fear response of the FO only group.

3. In Fig 1c and Supplementary Fig 5, ethanol significantly enhanced the fear response (recall+empathy) of the Priming+FO group whereas it had no effect on the fear response (empathy) of the FO only group. The simplest interpretation of this result may be that ethanol affects the recall-mediated, but not the empathy-related fear response.

4. In the same Fig1c, the suppressive effect of the oxytocin receptor antagonist on the ethanol-enhanced fear response of the Priming+FO group was not significant, although it seems to have induced an expansion of the variation in the direction of suppression. This could be due to the limited sample size used in the test. Or this may reflect the relatively small portion of the empathy fear component (sensitive to an oxytocin antagonist) compared to the big level of ethanol-enhanced, Priming-induced, fear (insensitive to an oxytocin antagonist).

5. A prior shock experience could induce a heightened state of anxiety as a result of a prior stressful experience. Although the authors showed no difference in anxiety in open field task, additional behavioral experiments may be needed to further clarify whether ethanol has any effects on other behavioral traits, such as anxiety-like traits in elevated-plus maze and conditioned fear in a classical fear conditioning.

6. Fig 2e, using overlap score for Arc mRNA localization both in Nuc+ and Cyto+, they demonstrated that the oxytocin receptor antagonist, L-368,899, reduced the overlap scores for neuronal activities in the Priming+FO group. The authors need to examine whether the increased ACC neuronal activities by ethanol injection is also affected by inhibition of oxytocin receptor. If the ethanol-enhanced fear is independent of oxytocin as the authors stated, one should see congruent results.

7. The authors demonstrated that the priming shock-induced behavioral enhancement was dependent on NMDA receptor activity. They also showed that prior shock experience preferentially activated BLA-projecting ACC pyramidal cells and ethanol increased activities of ACC neurons. In slice patch recording, the authors further attempted to explain this facilitatory effect by showing that ethanol decreased mEPSC in ACC pyramidal neurons, and suggested that the E/I imbalance in the ACC may cause the elevated fear in the Priming+FO mice. However, this explanation is not convincing because the diverse findings are not integrated to explain the final behavioral results. The authors need to address how the ethanol-induced increase in inhibitory synaptic transmission in the ACC contribute to the elevated fear.

8. In Fig 3 the authors showed that the overlap scores but not the percentage of Nuc+ cells in the ACC were positively correlated with the fear behavior. Why didn't they analyze the correlation between Cyt+ cells and fear behavior? This information seems critical considering that these cells are activated by the Priming.

9. The results on poly(I:C) autism mice in Fig 6 seem interesting. They, however, are too preliminary to give a solid conclusion regarding the underlying mechanism.

Reviewer #2 (Remarks to the Author):

This manuscript describes an extensive set of experiments designed to reveal the neural basis of a form of vicarious fear learning, as well as pharmacological modulators of this behavior. The study tackles an interesting and still much-understudied behavior (particularly when compared to the vast literature on direct fear conditioning), in the form of observational fear learning. It also benefits from the use of multiple approaches – behavioral, pharmacological, molecular (the painstaking FISH analysis of Arc-based ensemble-overlap results are especially laudable), and electrophysiological.

One concern with the current version of the manuscript – the many experiments and findings presented are not always well synthesized and the rationale for moving from one to the next is not always made clear. For example, the Introduction should end with a more specific set of hypothesis than 'whether drugs can modulate' – if this a study primarily about ethanol's actions on this behavior, as the title would indicate, then make this explicit. Another concern is that the mechanistic explanation of ethanol's effects in the ACC is a little underdeveloped: 1) the finding that ethanol increases net inhibition in cortex is not entirely new or unexpected, 2) it is unclear whether the decreases in mEPSC conductance is due to a change in mEPSC amplitude, frequency or duration, 3) are the effects on conductance AMPAR-mediated? Would the authors expect AMPAR manipulations to affect behavior? and 3) the benzo/GABA data provide indirect evidence of how ethanol is producing its effects – can the authors make a stronger case for a direct mechanistic connection?

Additional points:

- Why is behavior during conditioning only shown for the first experiment? These data should be provided for all experiments, even if supplemental only, to confirm the efficacy of the acquisition for each manipulation.
- How were drug doses chosen? Based on prior reports? If so, cite these sources in the methods. This is currently mentioned in the main text for ethanol only. In fact, there is a complete absence of Methods for the drug experiments (vehicles used, volumes injected, suppliers, etc).
- Acute ethanol has direct-fear memory-impairing effects at around the 2 g/kg dose in mice – please discuss the opposite effects reported here with observational fear – is this due to a lower dose used or a fundamental difference in the behavior/underlying mechanisms?

Reviewer #3 (Remarks to the Author):

The study by Sakaguchi et al. entitled, "Ethanol enhances empathy via recruitment of vicarious pain-sensitive neurons in the mouse anterior cingulate cortex", explores the effects of ethanol (EtOH) on observational fear following a footshock priming event. The authors demonstrate that priming increases socially-transmitted fear and this effect is diminished by an oxytocin receptor antagonist and enhanced by ethanol. They show using catFISH that a population of neurons in the anterior cingulate cortex (ACC) is activated by the priming event and fear observation. The oxytocin receptor antagonist decreases this overlap and EtOH increases it. They demonstrate a correlation between the overlap score and the percentage of defensive immobility. Using slice physiology, they then provide evidence that ethanol alters the excitation:inhibition ratio in the ACC. Finally, they show that EtOH rescues fear transmission in poly(I:C) mice.

This is an interesting study and the authors have done a nice job controlling their experiments and presenting their data as box plots. My main concerns are with their interpretation of some of the results. They need to clarify a number of issues with their results before I can recommend publication.

Major concerns:

- 1) "Empathy" in the title should be changed to either "empathy-like" or, even better, "observational fear" or "socially-transmitted fear". It is currently unclear whether rodents experience actual empathy.
- 2) The major crux of the paper is the effects of ethanol on socially transmitted fear. The data presented are all from the acquisition phase, during the fear observation period. Did the observer mice actually retain this learning? Do they exhibit defensive immobility if re-exposed to the conditioning context on the next day?
- 3) Previous work has shown that oxytocin decreases freezing responses via its effects on central amygdala circuits (see Viviani et al. 2011 and Knobloch et al. 2012 as examples). The authors should discuss why an oxytocin receptor antagonist decreases observational fear responses in their study.
- 4) The authors begin their paper with a discussion of the social bonding properties of alcohol. While this is true, alcohol also induces aggression in a subset of humans and rodents. How does this relate to the current study?
- 5) Were the data in this study tested for normality? Many of the Ns are quite low and it is likely that the authors should test for significance using nonparametric tests. All data should be tested for normality and the appropriate parametric or nonparametric test should be performed.

- 6) The naming of the priming only group as w/o shock is confusing because the mice are receiving a priming shock. The authors could clarify this by renaming this group as w/o FO for without Fear Observation.
- 7) Figure 1: is the defensive immobility of the w/o Priming group significantly higher than the control group? Why is there no effect of EtOH on defensive immobility of the w/o priming group? If EtOH increases empathy-like behavior, then it should increase defensive immobility in the w/o priming group.
- 8) Figure 2: Please show the overlap data for the priming and w/o priming groups. The statistics are reported but the values are not reported.
- 9) How did the authors determine that 7.0 ug/side ethanol was an appropriate dose for intra-ACC injection?
- 10) Figure 3: For panels b and c, which animals are included in this analysis? The N is reported to be 32. If mice from different treatment groups and experiments are included, the points should be color coded accordingly and a legend should be provided.
- 11) Figure 5: How is synaptic conductance calculated? The authors should also report the amplitude and frequency data.
- 12) Line 174: Why did the authors expect clonazepam, a drug that is frequently used to treat panic disorder, to increase fear transmission?
- 13) The authors report that there is no overlap in the BLA (line 146), but in Supplementary Figure 6, there is a significant difference in the % of double positive cells in the BLA. This should be clarified and discussed.
- 14) The Poly(I:C) mice are used in this study as a model for autism. I am surprised that the mice have normal fear learning and expression (Supplementary Figure 7), given the very high comorbidity of autism and anxiety disorders. The authors should clarify this point.

Comments and Answers

Reviewer #1

In recent years, the research into the biological basis of empathy has been expanded to animal models, and observational fear has been recognized as a useful model for assessing affective empathy in rodents. Observational fear is defined as social transfer of fear from one individual to another with social cues in the absence of direct aversive stimuli. Thus, mice can be conditioned for fear without receiving direct foot shocks.

In this manuscript, Sakaguchi et al., evaluated the contribution of alcohol to vicarious freezing in observational fear behavior that was primed by a prior foot shock experience. The authors found that ethanol strengthened observational fear response which was primed by a prior shock experience, whereas it did not affect the unprimed spontaneous observational fear response. An oxytocin receptor antagonist did not significantly affect the ethanol-enhanced freezing in the primed observational fear, whereas it reduced the freezing in the primed observational fear in the absence of ethanol. Next, they employed the Arc mRNA activity mapping, and identified that some ACC neurons were commonly activated by both the priming shock (Priming) and the fear observation (FO), and the degree of this overlap was positively correlated with the degree of freezing. In slice experiments on ACC pyramidal neurons, they further demonstrated that ethanol shifted the excitation/inhibition balance toward to inhibition. Experiments by utilizing Arc mRNA mapping combined with retrograde labelling showed that the ACC BLA projecting neurons were preferentially those activated by a priming shock (with/without overlap with fear observation) but not by fear observation alone. They also showed that ethanol injection increased observational fear in the poly(I:C)-induced autism mouse model. The authors concluded that alcohol enhances social transfer of fear in association with reduced E/I ratio in the ACC. Overall this work is interesting. However, there are major concerns in the design of the behavioral task, which limit the interpretation of the findings with regard to empathy.

We are grateful to this reviewer for her/his constructive comments, which have greatly improved our work. Individual responses are provided below:

1-1) In the behavioral paradigm using observer mice with a prior shock experience, the

authors found that a single priming shock (Priming) enhanced freezing of the observer mouse upon observation of a demonstrator receiving similar foot shocks (FO), and ethanol could further enhance this facilitating effect of the priming shock. Thus, the authors conclude that ethanol selectively strengthens the social transmission of fear. However, in their experimental protocol it should be noted that the observer mice experience two different sources of distress that may induce fear behavior: one through observation of demonstrator's distress without knowing the origin of the distress, and the other through remembering its own foot-shock experience on the grid (i.e., observation-induced recall of its own fear experience). While the former may involve social transmission of the affective state of the demonstrator mouse (empathy), the latter may primarily involve fear memory recall. Therefore, the results of ethanol experiments may mean that only the fear memory recall (but not empathy) is enhanced by ethanol (also stated below in 3). This perspective will inevitably change the scope of the work fundamentally.

We thank this reviewer for raising this important issue. We agree with this reviewer that it was possible that the defensive immobility in the PS+FO group included fear memory recall. Therefore, we have examined whether ethanol promotes recall of fear memory in a classical fear conditioning paradigm. We systemically injected 1.5 g/kg ethanol, the dose used in our study, 30 min before the recall test and found that ethanol significantly decreased, rather than increased, defensive immobility (Supplementary Fig. 7). This result suggests that ethanol suppresses context-dependent fear memory retrieval when applied before recall and is also consistent with previous reports showing that administration of similar doses of ethanol impairs memory retrieval in the Morris water maze task (1.5 g/kg, Chin et al., *Alcohol*, 2011) or the passive avoidance task (1.0 g/kg, Rostami et al., *Brain Res.*, 2017). Therefore, we believe that the facilitatory effect of ethanol on defensive immobility during the fear observation period did not result from ethanol-induced enhancement of associative memory recall. Moreover, we confirmed that our priming shock protocol *per se* did not enhance fear response (Fig. 1b) or induce contextual associative fear memory (Supplementary Fig. 1b, c). These results suggest that the immobility behavior observed in the PS+FO group and the fear memory recall in a classical fear conditioning are mediated, at least in part, through different neuronal mechanisms. However, as this reviewer states, ethanol did not increase immobility in the FO-only group (Supplementary Fig. 8c). Thus, it is also true that ethanol does not simply enhance fear transmission. Taken together, one of the most plausible interpretations is that ethanol enhances socially evoked fear memory recall. Indeed, human studies demonstrate that affective empathy consists of self-oriented affective responses and other-oriented responses (Batson et al., *J. Pers.*, 1987; Preis and Herwig,

Eur. J. Pain, 2012). These studies claim that "self-oriented affective responses include feelings of distress and anxiety that generate tendencies to eliminate one's own distress, whereas other-oriented responses comprise concern and sympathy which bring the well-being of the other person into focus, and both responses can occur together" (Batson et al., *J. Pers.*, 1987). This notion of affective empathy is not contradictory to our interpretation; that is, socially evoked fear memory recall can be regarded as a form of empathy. We have described these points in the Discussion section (P12, L274).

1-2) In Fig 1c, an oxytocin antagonist suppressed the fear response of the Priming+FO mice. Indeed, comparing the results between Fig1b and 1c, it is noted that the level of the remaining fear response after suppression by an oxytocin antagonist in the Priming+FO group seems similar to the difference in fear level between the Priming+FO group and the FO only group, revealing the sensitivity of the FO only group to an oxytocin antagonist. This result is consistent with the known role of oxytocin in empathy, and may exclude the priming-enhanced fear as a part of empathy behavior. The authors may want to confirm the effect of oxytocin antagonists on the fear response of the FO only group.

Thank you for this suggestion. We have performed additional experiments and found that the oxytocin receptor antagonist significantly decreased observational fear responses of the FO-only group (Supplementary Fig. 5). This suppressive effect was partial and was smaller than the effect observed in the PS+FO group (Fig. 1c). These results suggest that the priming-dependent fear component was also oxytocin-sensitive. Therefore, we cannot conclude that the priming-enhanced fear is not empathy at all. Indeed, in humans, the role of oxytocin in empathy is not simple. Because not all components of empathy are mediated by oxytocin signaling (Abu-Akel et al., *Soc. Neurosci.*, 2015; Leppanen et al., *Neurosci. Biobehav. Rev.*, 2017; Tully et al., *Psychiatry Res.*, 2018), the effectiveness of the oxytocin receptor antagonist alone cannot determine the involvement of empathy. We have provided additional information about oxytocin in the Introduction section (P4, L72).

1-3) In Fig 1c and Supplementary Fig 5, ethanol significantly enhanced the fear response (recall+empathy) of the Priming+FO group whereas it had no effect on the fear response (empathy) of the FO only group. The simplest interpretation of this result may be that ethanol affects the recall-mediated, but not the empathy-related fear response.

As we noted in comment #1-1, we do not exclude the possibility that the immobility behavior of the PS+FO group may contain the self-fear recall dimension, but we experimentally confirmed that ethanol does not enhance conditioned fear memory recall. We speculate that the main effect of ethanol is to enhance fear transmission or "socially evoked" fear memory recall, both of which are regarded as empathy-like behaviors.

1-4) In the same Fig1c, the suppressive effect of the oxytocin receptor antagonist on the ethanol-enhanced fear response of the Priming+FO group was not significant, although it seems to have induced an expansion of the variation in the direction of suppression. This could be due to the limited sample size used in the test. Or this may reflect the relatively small portion of the empathy fear component (sensitive to an oxytocin antagonist) compared to the big level of ethanol-enhanced, Priming-induced, fear (insensitive to an oxytocin antagonist).

We have increased the sample size in Fig. 1c from 12–13 mice to 22 mice per group. Ethanol-enhanced fear responses of the PS+FO group were still not significantly suppressed by the oxytocin receptor antagonist. However, as we mentioned in our answer to comment #1-2, the immobility behavior of the PS+FO group seems to be oxytocin-sensitive. Ethanol may decrease the contribution of oxytocin in inducing observational fear. We consider that ethanol may act downstream of oxytocin signaling. Oxytocin receptors are expressed mainly in inhibitory interneurons (Nakajima et al., *Cell*, 2014; Marlin et al., *Nature*, 2015), and oxytocin increases the firing rates of these neurons and thereby indirectly inhibits excitatory neurons (Owen et al., *Nature*, 2013; Nakajima et al., *Cell*, 2014; Oettl et al., *Neuron*, 2016). Ethanol may more directly modulate circuit excitability, independently of oxytocin receptors on the membrane surface of inhibitory neurons.

1-5) A prior shock experience could induce a heightened state of anxiety as a result of a prior stressful experience. Although the authors showed no difference in anxiety in open field task, additional behavioral experiments may be needed to further clarify whether ethanol has any effects on other behavioral traits, such as anxiety-like traits in elevated-plus maze and conditioned fear in a classical fear conditioning.

We appreciate this valuable comment. In response to this suggestion, we have conducted the elevated plus-maze task and classical fear conditioning (the latter described in comment #1-1) in addition to the open field test. In the open field test,

ethanol increased the total distance traveled, but it did not alter the immobility time or time spent in the center area (Supplementary Fig. 6a–d). These results were not affected by a prior shock experience. In the elevated plus-maze task, ethanol increased the number of open arm entries only when the mice had received a priming shock, whereas it did not change time spent in the open arms (Supplementary Fig. 6e,f). These data suggest that ethanol increases locomotor activity and tends to reduce the anxiety level. In classical conditioning tasks, as we wrote in our answer #1-1, ethanol significantly decreased defensive immobility during the test period (Supplementary Fig. 7). Thus, ethanol attenuates retrieval of contextual associative memory. These general effects of ethanol cannot explain our finding that ethanol enhances the defensive immobility during the fear observation. We described these results in the revised manuscript (P6, L129).

1-6) Fig 2e, using overlap score for Arc mRNA localization both in Nuc+ and Cyto+, they demonstrated that the oxytocin receptor antagonist, L-368,899, reduced the overlap scores for neuronal activities in the Priming+FO group. The authors need to examine whether the increased ACC neuronal activities by ethanol injection is also affected by inhibition of oxytocin receptor. If the ethanol-enhanced fear is independent of oxytocin as the authors stated, one should see congruent results.

According to this suggestion, we have made additional experiments to analyze *Arc* mRNA expression in the PS+FO group that was co-treated with ethanol and L-368,899 (Fig. 2e, EtOH+L-368,899 group). We found that the overlap score in the EtOH+L-368,899 group did not differ significantly from that in the EtOH group, suggesting that the ethanol-induced increase in ACC population overlap was not mediated by oxytocin receptor activity. Thus, this result appears consistent with the behavioral data in Fig. 1c. This point is described in the Results section (P7, L163).

1-7) The authors demonstrated that the priming shock-induced behavioral enhancement was dependent on NMDA receptor activity. They also showed that prior shock experience preferentially activated BLA-projecting ACC pyramidal cells and ethanol increased activities of ACC neurons. In slice patch recording, the authors further attempted to explain this facilitatory effect by showing that ethanol decreased mEPSC in ACC pyramidal neurons, and suggested that the E/I imbalance in the ACC may cause the elevated fear in the Priming+FO mice. However, this explanation is not convincing because the diverse findings are not integrated to explain the final behavioral results.

The authors need to address how the ethanol-induced increase in inhibitory synaptic transmission in the ACC contribute to the elevated fear.

We agree with this reviewer that our results are somewhat fragmented and do not necessarily explain the causal relationships. As the background, we focused on the relationship between the E/I balance and neuronal population overlap, which was recently revealed in the research field of memory (Cai et al., *Nature*, 2016, Yokose et al., *Science*, 2017). The E/I balance in individual neurons is a fundamental factor in memory allocation (Mizunuma et al., *Nat. Neurosci.*, 2014). Indeed, neurons are recruited to a memory trace based on relative neuronal excitability immediately before training (Yiu et al., *Neuron*, 2014). These facts suggest that overlapping neuronal representations are shaped by neuronal E/I balance. Recently, Rashid et al. approached this hypothesis by bidirectionally manipulating neuronal excitability (*Science*, 2016). They found that two independent learning events recruited partially overlapping neuronal populations in the lateral amygdala, and this overlap was reduced by chemogenetic suppression of PV-expressing interneurons. They further demonstrated that PV interneurons suppress non-allocated principal neurons after learning. These results suggest that the E/I balance determines whether two memories are bound or segregated by modulating the overlap between the two memory engrams. Our study expands this notion to a domain of inter-individual experiences and showed that the overlap between the neuronal representations for experienced and observed pain was increased by ethanol, which shifts the E/I balance toward inhibition. We also showed that the priming shock preferentially activated BLA-projecting ACC neurons. This tendency was not found in the fear observation. These results imply that the increased overlap of ACC neuron ensembles reflects increased functional connectivity between the ACC and BLA, which potentially contributes to effective emotional transition during the fear observation. In the Discussion part of the revised manuscript, we have re-provided the background information and hypotheses and tried to link these apparently disparate data (P11, L245).

1-8) In Fig 3 the authors showed that the overlap scores but not the percentage of Nuc⁺ cells in the ACC were positively correlated with the fear behavior. Why didn't they analyze the correlation between Cyt⁺ cells and fear behavior? This information seems critical considering that these cells are activated by the Priming.

Thank you for this suggestion. We have added the correlation plot for defensive immobility *versus* the percentage of Cyt⁺ cells (Fig. 3), which shows no significant

correlation. Thus, the sizes of neuron populations activated by the priming shock were unlikely to affect the subsequent fear transmission.

1-9) The results on poly(I:C) autism mice in Fig 6 seem interesting. They, however, are too preliminary to give a solid conclusion regarding the underlying mechanism.

We agree with this reviewer that the data from the poly(I:C) mice fail to provide the underlying mechanisms. Nonetheless, we believe that these data still contribute to our interpretation of the role of the E/I balance in sociality. While mouse models of ASD are generally evaluated by their reduced interaction time in the three-chamber test, the properties of empathy-relevant behaviors in ASD model mice are largely unknown. To our knowledge, our study is the first to show the impairment of social fear transmission in ASD model mice and the related neuronal activity at the single cell level. Moreover, while the imbalanced E/I ratios are widely observed in ASD model mice, the mechanisms linking the E/I imbalance and social impairment remain unclear. Our findings, namely, that ethanol enhances fear transmission in poly(I:C) mice to a level comparable to that in control mice by reducing the E/I ratio and by increasing the shared neuronal representations for experienced and observed pain, provide insights into our understanding of the relationship between the neocortical E/I balance and sociality. Moreover, we showed that the reduced fear transmission in poly(I:C) mice cannot be explained by learning capability. This fact is consistent with our interpretation that observational fear enhanced by a priming shock is not merely derived from the memory retrieval.

Reviewer #2

This manuscript describes an extensive set of experiments designed to reveal the neural basis of a form of vicarious fear learning, as well as pharmacological modulators of this behavior. The study tackles an interesting and still much-understudied behavior (particularly when compared to the vast literature on direct fear conditioning), in the form of observational fear learning. It also benefits from the use of multiple approaches – behavioral, pharmacological, molecular (the painstaking FISH analysis of Arc-based ensemble-overlap results are especially laudable), and electrophysiological.

Thank you for the positive evaluations, which have encouraged us to resubmit this manuscript. We have revised it in accordance with these comments. Our point-by-point responses are as follows:

2-1) One concern with the current version of the manuscript – the many experiments and findings presented are not always well synthesized and the rationale for moving from one to the next is not always made clear. For example, the Introduction should end with a more specific set of hypothesis than ‘whether drugs can modulate’ – if this a study primarily about ethanol’s actions on this behavior, as the title would indicate, then make this explicit.

We thank you for this comment. According to this suggestion, we have re-written the corresponding part in the Introduction (P4, L70) as "whether ethanol modulates the overlapping ACC neuronal representation and observational fear", which points directly to the main hypothesis of our study. We have also carefully revised the entire manuscript to make the relationships between individual experiments more logical and rational.

2-2-i) Another concern is that the mechanistic explanation of ethanol’s effects in the ACC is a little underdeveloped: The finding that ethanol increases net inhibition in cortex is not entirely new or unexpected.

As this reviewer claims, it is widely assumed that ethanol increases net inhibition in the central nervous system. However, the underlying mechanisms vary considerably among brain regions and also among neuronal subtypes. For example, ethanol increases the amplitude of IPSCs in the central nucleus of the amygdala (Roberto et al., *PNAS*, 2003) and in the hippocampal CA1 (Weiner et al., *J. Neurochem.*, 1997). In contrast, in the cerebellar Purkinje neurons, ethanol increases only the IPSC frequency without changes in the amplitude or the decay time (Ming et al., *Alcohol Clin. Exp. Res.*, 2006), probably due to an increase in the firing rates of interneurons (Hirono et al., *Neuropharmacology*, 2009). Another study reported that ethanol decreases AMPAR-mediated currents in the somatosensory cortex (Lu and Yeh, *Neurochem. Int.*, 1999). Thus, we needed to evaluate the effect of ethanol on ACC neurons in our system.

2-2-ii) It is unclear whether the decreases in mEPSC conductance is due to a change in

mEPSC amplitude, frequency or duration.

Thank you for raising this point. We have presented the data of the amplitudes and frequencies of mEPSCs and mIPSCs in Supplementary Fig. 10. These data indicate that the ethanol-induced decrease in the mEPSC conductance was due to decreases in both the amplitudes and frequencies. This is explained in the Results section (P9, L194). We did not calculate the duration because multiple PSC events sometimes occurred at very short intervals and were difficult to resolve into individual durations. Indeed, this was the reason why we employed the conductances for data comparison.

2-2-iii) Are the effects on conductance AMPAR-mediated? Would the authors expect AMPAR manipulations to affect behavior?

We have confirmed that mEPSCs in ACC pyramidal neurons are abolished by bath application of DNQX, an AMPAR antagonist (Supplementary Fig. 10a). Based on these data, we think that AMPAR manipulation, like our GABAR manipulation, may affect social fear transmission.

2-2-iv) The benzo/GABA data provide indirect evidence of how ethanol is producing its effects – can the authors make a stronger case for a direct mechanistic connection?

Thank you for this comment. To provide more direct evidence, we previously tried to conduct chemogenetic manipulations of ACC neuronal activity to examine how fear transmission and the effect of ethanol on it were influenced, but we were not able to isolate the effect of the chemogenetic manipulations, which nonspecifically affected the behaviors of mice due to unknown reasons (perhaps empathy is supported by remarkably delicate behavior/emotion?). However, conceptually, this comment is related to comment #1-7. In Fig. 5a,b, we showed that ethanol decreased the mEPSC conductance and increased the mIPSC conductance, and thus, shifted synaptic E/I balance toward inhibition. Therefore, we tested whether this balance shift is important for the facilitatory effect of ethanol on fear transmission. In the experiment in Fig. 5c, we have shown that the behavioral effect of ethanol was blocked by intra-ACC injection of a GABA_A receptor antagonist, picrotoxin. This result suggests that the effect of ethanol requires a shift in the E/I balance in the ACC. On the other hand, we have also shown that ethanol increased the overlap between neuronal representation of self-experienced and observed pain with *Arc* catFISH experiment (Fig. 2e). As we

described in response to comment #1-7, recent studies about learning and memory suggest that shifting the E/I balance toward inhibition leads to an increase in reactivation of a neuron population that was activated by a past event (Rashid et al., *Science*, 2016). Although we did not show a direct causal relationship between enhanced fear transmission and the population overlap, the shifted E/I ratio may represent a mechanism underlying the facilitatory effect of ethanol on fear transmission.

2-3) Why is behavior during conditioning only shown for the first experiment? These data should be provided for all experiments, even if supplemental only, to confirm the efficacy of the acquisition for each manipulation.

We apologize for our unclear writing. Virtually all data that we labeled as "defensive immobility" represent immobile behavior 'during the observation periods', except for Supplementary Figs. 1c, 7b and 11b, which show the data in the fear conditioning experiments. These exception data are indicated as "2-h Test" or "24-h Test" above each graph. Please note that our study did not aim to examine conditioning (*i.e.*, acquisition or retrieval of fear) but wanted to measure instantaneous fear expression while mice observed other's pain for the first time.

2-4) How were drug doses chosen? Based on prior reports? If so, cite these sources in the methods. This is currently mentioned in the main text for ethanol only. In fact, there is a complete absence of Methods for the drug experiments (vehicles used, volumes injected, suppliers, etc).

We apologize for the insufficient explanation. The doses of all drugs were chosen based on previous reports from our and other laboratories. For all drugs, we have cited the reports we referred to. As vehicles, we used 0.9% NaCl (saline) except for clonazepam, for which we used saline with 0.1% DMSO. The injected volumes were 10 ml/kg for systemic injection and 0.5 μ l/side for intra-ACC injection. We have added these details in the Methods section (P16, L366).

2-5) Acute ethanol has direct-fear memory-impairing effects at around the 2 g/kg dose in mice – please discuss the opposite effects reported here with observational fear – is this due to a lower dose used or a fundamental difference in the behavior/underlying mechanisms?

We appreciate you for raising this important issue. This comment is also related to comments #1-1 and #1-3. Indeed, we found that acute 1.5 g/kg ethanol impairs contextual fear memory recall (Supplementary Fig. 7). We think that this fact does not conflict with our data of empathy because the underlying mechanisms may be different. In classical fear conditioning, electrical shocks are associated with neutral contexts, and fear responses are induced by re-exposure to the originally neutral context. On the other hand, in our fear observation paradigm, fear responses were instantaneously induced by the behavior of a demonstrator (Fig. 1b), and thus, the fear observation is not mediated by typical associational learning. Moreover, we confirmed that our priming shock protocol was insufficient to form associative memory (Supplementary Fig. 1c), suggesting that the immobility in the PS+FO group is not dependent on associative memory recall. Taken together, the memory recall in classical fear conditioning and observational fear responses (following a priming shock) forms, at least in part, through different mechanisms. Therefore, we consider that it is not strange even if the effects of ethanol were apparently opposite in these two behavioral paradigms.

Reviewer #3

The study by Sakaguchi et al. entitled, “Ethanol enhances empathy via recruitment of vicarious pain-sensitive neurons in the mouse anterior cingulate cortex”, explores the effects of ethanol (EtOH) on observational fear following a footshock priming event. The authors demonstrate that priming increases socially-transmitted fear and this effect is diminished by an oxytocin receptor antagonist and enhanced by ethanol. They show using catFISH that a population of neurons in the anterior cingulate cortex (ACC) is activated by the priming event and fear observation. The oxytocin receptor antagonist decreases this overlap and EtOH increases it. They demonstrate a correlation between the overlap score and the percentage of defensive immobility. Using slice physiology, they then provide evidence that ethanol alters the excitation:inhibition ratio in the ACC. Finally, they show that EtOH rescues fear transmission in poly(I:C) mice.

This is an interesting study and the authors have done a nice job controlling their experiments and presenting their data as box plots. My main concerns are with their interpretation of some of the results. They need to clarify a number of issues with their results before I can recommend publication.

We appreciate that this reviewer found scientific value in our manuscript. Thanks to the comments, we are pleased to have been able to revise it into a better manuscript. Individual responses are listed below:

3-1) “Empathy” in the title should be changed to either “empathy-like” or, even better, “observational fear” or “socially-transmitted fear”. It is currently unclear whether rodents experience actual empathy.

Following this suggestion, we have changed the term "empathy" in the title to "empathy-like behaviors".

3-2) The major crux of the paper is the effects of ethanol on socially transmitted fear. The data presented are all from the acquisition phase, during the fear observation period. Did the observer mice actually retain this learning? Do they exhibit defensive immobility if re-exposed to the conditioning context on the next day?

Thank you very much for raising this question. We have conducted additional experiments to examine whether the enhancing effect persists. We found that the observer mice showed immobility behavior when they were re-exposed to the conditioned context 24 h after the fear observation (Supplementary Fig. 1), suggesting that they learned fear through observation of fear expressed by their cagemate. Importantly, the priming shock alone was insufficient to induce this conditioning. These results have been described in the Results section (P5, L100). Moreover, we also examined the effect of ethanol on this fear-observation conditioning. Injection of ethanol 30 min before the fear observation enhanced fear responses only during the observation periods, not during the test period on the next day. This result suggests that ethanol did not facilitate memory acquisition even though it instantaneously enhanced observatory fear expression.

3-3) Previous work has shown that oxytocin decreases freezing responses via its effects on central amygdala circuits (see Viviani et al. 2011 and Knobloch et al. 2012 as examples). The authors should discuss why an oxytocin receptor antagonist decreases observational fear responses in their study.

Thank you for this important comment. As this reviewer describes, several reports have shown that oxytocin attenuates fear through modulating the central amygdala activity (Viviani et al., *Science*, 2011; Knobloch et al., *Neuron*, 2012). However, these studies only showed that artificially elevated oxytocin signaling (*i.e.*, microinjection of an oxytocin receptor agonist or ChR2-evoked axonal oxytocin release) in the central amygdala caused fear reduction. They did not mention how strongly natural oxytocin release is involved in contextual freezing behavior. More recently, Pisansky et al. showed that systemic administration of an oxytocin receptor antagonist (L-368,899, 5 or 10 mg/kg) did not alter fear response in a Pavlovian conditioning (Pisansky et al., *Nat. Commun.*, 2017). Consistent with this recent literature, a meta-analysis of human studies found that intranasal oxytocin did not significantly influence the expression of negative emotions, including fear (Leppanen et al., *Neurosci. Biobehav. Rev.*, 2017). Therefore, it is unclear whether "naturally released" oxytocin in the amygdala is truly involved in fear conditioning. On the other hand, we showed that intra-ACC injection of L-368,899 is sufficient to reduce fear transmission (Fig. 3a), suggesting that oxytocin has different effects on fear in a Pavlovian conditioning task and socially transmitted fear. This difference may also simply reflect the difference in the brain regions. Another possible interpretation is that oxytocin may, in fact, attenuate fear, but this suppressive effect can be surpassed by oxytocin-induced facilitation of social fear transmission. We described these possibilities in the Discussion section (P14, L324).

3-4) The authors begin their paper with a discussion of the social bonding properties of alcohol. While this is true, alcohol also induces aggression in a subset of humans and rodents. How does this relate to the current study?

Thank you for this comment. Indeed, several studies reported that ethanol can enhance aggression in mice (Faccidomo et al., *Neuropsychopharmacology*, 2008; Quadros et al., *Neuropsychopharmacology*, 2014; Newman et al., *Psychopharmacology*, 2012), but we believe that our data do not conflict with these studies. All these studies used the so-called resident-intruder paradigm, in which the resident mice attack unfamiliar mice that intrude on their territory (Koolhaas et al., *J. Vis. Exp.*, 2013). This situation is very different from the conditions of our study; the observer and the demonstrator were

assigned from co-housed mice, and thus, they essentially do not exhibit offensive behavior against one another. The familiarity between the two animals is known as a crucial factor in determining empathy-related behaviors, such as fear transmission, pro-social behavior, and social modulation of pain (Jeon et al., *Nat. Neurosci.*, 2010; Bartal et al., *eLIFE*, 2014; Langford et al., *Science*, 2006). For example, oxytocin is likely to augment in-group favoritism and out-group derogation (De Dreu et al., *Proc. Natl. Acad. Sci. U. S. A.*, 2011). Ethanol may work in a similar way. This has been described in the Discussion section (P14, L314).

3-5) Were the data in this study tested for normality? Many of the Ns are quite low and it is likely that the authors should test for significance using nonparametric tests. All data should be tested for normality and the appropriate parametric or nonparametric test should be performed.

Thank you for this valuable comment. According to this suggestion, all data have been tested for normality using the Shapiro-Wilk test, and the statistical comparisons have been fully revised. Some data were judged not to conform to normal distributions; therefore, we used a bootstrap resampling method that does not require normality. Nevertheless, most of the results of statistical tests were not changed. The exceptions were Figs. 5b *Middle*, S9, and S11b *Right*, but these changes did not affect our conclusions or interpretations of other data (please see also our answer to comment #3-14).

3-6) The naming of the priming only group as w/o shock is confusing because the mice are receiving a priming shock. The authors could clarify this by renaming this group as w/o FO for without Fear Observation.

We apologize for causing this confusion. We have renamed the previous "w/o Shock" group with Priming Shock only as "PS-only". We have also re-labeled the "w/o Priming" group as "FO-only" and the "Priming" group as "PS+FO", respectively, for simplicity.

3-7) Figure 1: is the defensive immobility of the w/o Priming group significantly higher than the control group? Why is there no effect of EtOH on defensive immobility of the

w/o priming group? If EtOH increases empathy-like behavior, then it should increase defensive immobility in the w/o priming group.

Yes, the defensive immobility in the FO-only group was significantly higher than in the control group. This statistical significance is now indicated in the graph and the legend in Fig. 1b. Ethanol enhanced fear transmission in the PS+FO group but not in the FO-only group. This selectivity is consistent with our catFISH data, which suggest that ethanol enhanced fear transmission through an increased overlap of neuronal representations between experienced and observed pain. As we noted in response to comment #1-7, we believe that defensive immobility expressed by not only the FO-only group but also the PS+FO group represents a component of affective empathy.

3-8) Figure 2: Please show the overlap data for the priming and w/o priming groups. The statistics are reported but the values are not reported.

According to this suggestion, we have added the overlap scores for the PS+FO and FO-only groups in the main text (P7, L162).

3-9) How did the authors determine that 7.0 ug/side ethanol was an appropriate dose for intra-ACC injection?

We referred to previous reports about rodent behaviors (including a study by our laboratory) to determine the dose of ethanol for intra-ACC perfusion (Mao et al., *Neurosci. Lett.*, 1996; Nomura and Matsuki, *Neuropsychopharmacol.*, 2008). We also referred to the estimated concentrations in the brain after systemic injection of ethanol (Ferraro et al., *Alcohol*, 1990).

3-10) Figure 3: For panels b and c, which animals are included in this analysis? The N is reported to be 32. If mice from different treatment groups and experiments are included, the points should be color coded accordingly and a legend should be provided.

We apologize for the insufficient explanation. For these graphs, we pooled data in the PS+FO group in Fig. 2c, drug-treated groups (saline, L-368,899, EtOH) in Fig. 2e and the vehicle-treated group (control for poly(I:C)) in Fig. 6c. As this reviewer suggested, we have re-drawn the graphs and rewritten the legend to clarify the animal groups

analyzed.

3-11) Figure 5: How is synaptic conductance calculated? The authors should also report the amplitude and frequency data.

We calculated the synaptic conductance by dividing the charge (the time integral of excitatory/inhibitory postsynaptic currents) by the driving force (clamped voltage) (Mizunuma et al., *Nat. Neurosci.*, 2014). This measure is believed to reflect general synaptic efficacy, including the amplitudes, durations, and frequencies of synaptic events (Gardner, *J. Physiol.*, 1980). This explanation now appears in the Methods section (P22, L512). Moreover, we have displayed the amplitude and frequency data in Supplementary Fig. 10. These data may help readers to understand the synaptic mechanisms underlying the action of ethanol.

3-12) Line 174: Why did the authors expect clonazepam, a drug that is frequently used to treat panic disorder, to increase fear transmission?

Because our electrophysiological data suggested that ethanol reduces the synaptic E/I ratio and thereby enhances fear transmission, we predicted that GABA-enhancing drugs, such as benzodiazepines, would also show a similar effect at the behavioral level. Among benzodiazepines, clonazepam is reported to reduce the synaptic E/I ratio and thereby ameliorate behavioral deficits in a mouse model of autism (Han et al., *Neuron*, 2014). Thus, we reasoned that clonazepam could increase socially transmitted fear. We have added this explanation in the revised manuscript (P9, L201).

3-13) The authors report that there is no overlap in the BLA (line 146), but in Supplementary Figure 6, there is a significant difference in the % of double positive cells in the BLA. This should be clarified and discussed.

We evaluated the overlap between two neuronal populations by using the "overlap score", which is defined such that it excludes the effects of population sizes (Vazdarjanova and Guzowski, *J. Neurosci.*, 2004). Indeed, this overlap score, when based on *Arc* gene expression patterns, is reported to behave independently of the population size (Nomura et al., *NeuroImage*, 2012). Thus, it is mathematically possible for there to have been a significant difference in the percentage of double-positive cells

in the BLA without affecting the overlap score. In our study, we focused on the ACC because the ACC was the only brain region that exhibited a significantly higher overlap score in the PS+FO group. Of course, we think that the population sizes may also be important. In particular, the ACC has functional connections with the BLA and the AIC, both of which exhibited high proportions of double-positive cells. We believe that this finding provides an interesting direction for a future study. To avoid confusion, we changed the sentence "Although there was no evidence for a significant overlap in the BLA" in the Results section to "Although the overlap score in the BLA did not differ between the PS-only and PS+FO groups" (P8, L177).

3-14) The Poly(I:C) mice are used in this study as a model for autism. I am surprised that the mice have normal fear learning and expression (Supplementary Figure 7), given the very high comorbidity of autism and anxiety disorders. The authors should clarify this point.

First of all, we have re-analyzed these data using a bootstrap test because their distributions did not pass the Shapiro-Wilk test for normality. Our new statistical analysis reveals that the immobility time in the recall test 24 h after the conditioning was significantly higher in the poly(I:C)-treated mice than in the vehicle-treated mice (Supplementary Fig. 11b, *Right*). As this reviewer expected, this difference may reflect an increased anxiety level, which is frequently observed in autism model rodents (Patterson, *Pediatr. Res.*, 2011). However, the immobility time in the recall test 2 h after the conditioning did not differ between the two groups (Supplementary Fig. 11b, *Middle*). This lack of a difference may be due to the ceiling effect; please note that the immobility time was as high as approximately 80% even in the vehicle group at this time point.

Reviewers' comments:

Reviewer #1 (Remarks to the Author):

The revised MS is much improved and the authors addressed some of the issues raised. Overall, the main output of the paper, showing that ethanol facilitates a priming-shock coupled observational fear response, provides an interesting addition to social behavioral studies using the observational fear paradigm. However, the major issue, defining the nature of the behavior they are looking at, is still there. The authors' rebuttal to the reviewer's comment does not successfully support their conclusion. Thus, the authors' interpretation of their results in the context of empathy is not supported with the findings, while there are multiple findings against it. I will try to highlight the problems below.

1. Affective/emotional empathy is broadly defined as emotional state-matching between individuals. In observational fear, a subject ('observer') expresses a change in emotion in response to an affective experience of a conspecific ('demonstrator') in the absence of direct aversive stimuli to itself. In other words, what is transmitted from the demonstrator to the observer is information related to the affective state: the core element of affective empathy. This is the reason why the term, affective empathy, has been used to describe this phenomenon. It is well known that a preceding experience of the observer of the similar aversive stimuli that the demonstrator receives can enhance the observational fear response of the observer animal. This is the case for the fear response in the PS+FO group. It is important to note that this observational fear response should contain the portion induced by the socially evoked fear memory recall (in the authors' words) in addition to the empathy-related fear. On the other hand, the vicarious freezing in the FO mice was evoked solely by socially transmitted fear without a priming shock, and this has been interpreted in the field to be based on affective empathy (Chen et al., 2009; Jeon et al., 2010; Gonzalez-Liencre et al., 2014; Lidhar et al., 2017; Debiec and Olsson 2017; Pisansky et al., 2017). The authors claim that socially evoked fear memory recall is also empathy is not substantiated. It does not contain the core element of affective empathy.

It would be a very interesting and exciting project to figure out how the socially evoked fear memory recall leads to enhanced fear response in the PS+FO group in the brain. That must be different, however, from the neural mechanism underlying affective empathy, taking into consideration the experimental results provided in the manuscript, including 1) it is not influenced by the oxytocin system and 2) it is enhanced by ethanol. These findings seem to distinguish the two different events leading to fear. Apparently it must also be different from that underlying the classical associative memory as noted by the authors.

Now, in this study, ethanol showed no effect on the fear response of the FO group, and the facilitating effect of ethanol was seen only when combined with a priming shock, PS+FO. The simplest interpretation of these results would be that ethanol affects the behavior related to socially evoked fear memory recall, but not the empathy-related fear response. In fact the authors' statement says the same thing: [Thus, it is also true that ethanol does not simply enhance fear transmission. Taken together, one of the most plausible interpretations is that ethanol enhances socially evoked fear memory recall.] It should not be ignored that

fear transmission is the core of affective empathy.

2. Authors' citation and discussion of the work by (Batson et al., J. Pers., 1987; Preis and Herwig, Eur. J. Pain, 2012), I am afraid, seem out of the focus. My understanding is that the cited works tried to categorize affective empathy based on behavioral implications, not on how it is acquired. The fear response of the FO group must also contain, if not solely, self-oriented affective responses. Following the authors' logic, ethanol should affect this fear response, too. But it did not.

3. The authors' explanation of how the shifted E/I balance by ethanol is related to the enhanced fear behavior is still not convincing.

The authors showed that the behavioral effect of ethanol was blocked by intra-ACC injection of the GABA-A receptor blocker, picrotoxin, and thus claimed that ethanol-mediated E/I-balance shift to inhibition in the ACC increased observational fear. However, in Figure 5c, injection of picrotoxin alone in the absence of ethanol administration almost completely abolished the observational fear response. Do the authors think this shifted E/I balance toward excitation suppresses ethanol-enhanced observational fear? The experimental design is too crude to allow such an elegant conclusion. Could a simpler explanation be that the impaired observational fear response in the picrotoxin+EtOH group might be just due to dysfunctional neuronal circuits in the ACC induced by picrotoxin?

Reviewer #2 (Remarks to the Author):

Remaining comments:

- Please refer to and cite the intranasal oxytocin paper in the Introduction.
- Note the duration of co-housing in the opening lines of the Results – this is a critical variable.
- 'neighboring compartment' not neighboring room' in the first paragraph of the Results.
- Please note where the hippocampal CA1 sampling was done – dorsal or ventral?
- Statistical results are sometimes presented in the Results and other times (mainly) the Figure Legends – please be consistent.
- Add the lux level used in the open field and elevated plus-maze.
- Note on page 20 when precisely mice were killed post testing.
- For Figure 4 BLA CTb injections – show an example of the injection site expression.

Reviewer #3 (Remarks to the Author):

The revised manuscript by Dr Ikegaya and colleagues is greatly improved. The authors worked very hard to address my concerns, and the additional experiments, analyses, revamped statistics, and updated text make the paper better. I have only one remaining concern. The fact that ethanol does not enhance defensive immobility in the FO-only group does not support their general assertion that ethanol enhances empathy-like behavior. The authors have attempted to explain this using the argument that there is a combined

memory recall and fear transmission effect. I think this is an acceptable argument, but the manuscript language needs to be tightened throughout to reflect this interpretation. For example, the abstract states that ethanol, "facilitates observational fear transmission" (Line 32). This and other statements need to be changed to reflect their updated hypothesis so as to not be misleading. Possibilities could include using "socially evoked fear memory recall" or "priming-induced fear transmission". If the authors update the text of the manuscript to reflect their updated hypothesis, I am happy to recommend publication.

Reviewer #1

1-1) The revised MS is much improved and the authors addressed some of the issues raised. Overall, the main output of the paper, showing that ethanol facilitates a priming-shock coupled observational fear response, provides an interesting addition to social behavioral studies using the observational fear paradigm. However, the major issue, defining the nature of the behavior they are looking at, is still there. The authors' rebuttal to the reviewer's comment does not successfully support their conclusion. Thus, the authors' interpretation of their results in the context of empathy is not supported with the findings, while there are multiple findings against it. I will try to highlight the problems below.

Affective/emotional empathy is broadly defined as emotional state-matching between individuals. In observational fear, a subject ('observer') expresses a change in emotion in response to an affective experience of a conspecific ('demonstrator') in the absence of direct aversive stimuli to itself. In other words, what is transmitted from the demonstrator to the observer is information related to the affective state: the core element of affective empathy. This is the reason why the term, affective empathy, has been used to describe this phenomenon. It is well known that a preceding experience of the observer of the similar aversive stimuli that the demonstrator receives can enhance the observational fear response of the observer animal. This is the case for the fear response in the PS+FO group. It is important to note that this observational fear response should contain the portion induced by the socially evoked fear memory recall (in the authors' words) in addition to the empathy-related fear. On the other hand, the vicarious freezing in the FO mice was evoked solely by socially transmitted fear without a priming shock, and this has been interpreted in the field to be based on affective empathy (Chen et al., 2009; Jeon et al., 2010; Gonzalez-Liencre et al., 2014; Lidhar et al., 2017; Debiec and Olsson 2017; Pisansky et al., 2017). The authors claim that socially evoked fear memory recall is also empathy is not substantiated. It does not contain the core element of affective empathy.

It would be a very interesting and exciting project to figure out how the socially evoked fear memory recall leads to enhanced fear response in the PS+FO group in the brain. That must be different, however, from the neural mechanism

underlying affective empathy, taking into consideration the experimental results provided in the manuscript, including 1) it is not influenced by the oxytocin system and 2) it is enhanced by ethanol. These findings seem to distinguish the two different events leading to fear. Apparently it must also be different from that underlying the classical associative memory as noted by the authors.

Now, in this study, ethanol showed no effect on the fear response of the FO group, and the facilitating effect of ethanol was seen only when combined with a priming shock, PS+FO. The simplest interpretation of these results would be that ethanol affects the behavior related to socially evoked fear memory recall, but not the empathy-related fear response. In fact the authors' statement says the same thing: [Thus, it is also true that ethanol does not simply enhance fear transmission. Taken together, one of the most plausible interpretations is that ethanol enhances socially evoked fear memory recall. It should not be ignored that fear transmission is the core of affective empathy.

Thank you for your very attentive comment. There are the long history of research regarding empathy in the field of psychology, through which various definitions of empathy have been proposed. In response to this comment, all the authors, who initially had different opinions on this issue, gathered several times and carefully and intensively discussed our stance and our data interpretations. After long discussion, the corresponding author was convinced and changed his mind. We all have now agreed that this reviewer's claim is the most standard and reflects common-sense definition of affective empathy. We have also agreed that the observational fear response in the FO-only group and the PS+FO group are qualitatively different, given the results of sensitivity to the pharmacological manipulations. As our consensus, we have decided to change the expressions for the priming shock-coupled observational fear response and have regarded the effect of ethanol as the facilitation of "socially evoked fear memory recall".

1-2) Authors' citation and discussion of the work by (Batson et al., J. Pers., 1987; Preis and Herwig, Eur. J. Pain, 2012), I am afraid, seem out of the focus. My understanding is that the cited works tried to categorize affective empathy based on behavioral implications, not on how it is acquired. The fear response of the FO group must also contain, if not solely, self-oriented affective responses. Following the authors' logic, ethanol should affect this fear response, too. But it

did not.

We appreciate you for this valuable comment. We also agree with this reviewer that the fear response of the FO-only group may contain self-oriented affective elements, although we cannot conclude it only from our behavioral observations. As we described in the answer to comment #1-1, we have decided to change our interpretation for the relationship between our behavioral data and affective empathy, and have carefully rewritten the corresponding parts of the manuscript to more appropriately describe the behavioral implications in our experimental paradigm.

1-3) The authors' explanation of how the shifted E/I balance by ethanol is related to the enhanced fear behavior is still not convincing. The authors showed that the behavioral effect of ethanol was blocked by intra-ACC injection of the GABA-A receptor blocker, picrotoxin, and thus claimed that ethanol-mediated E/I-balance shift to inhibition in the ACC increased observational fear. However, in Figure 5c, injection of picrotoxin alone in the absence of ethanol administration almost completely abolished the observational fear response. Do the authors think this shifted E/I balance toward excitation suppresses ethanol-enhanced observational fear? The experimental design is too crude to allow such an elegant conclusion. Could a simpler explanation be that the impaired observational fear response in the picrotoxin+EtOH group might be just due to dysfunctional neuronal circuits in the ACC induced by picrotoxin?

Thank you for raising this important issue. Yes, our pharmacological data alone cannot necessarily substantiate that ethanol-induced E/I-balance shift toward inhibition in the ACC mediates ethanol-facilitated observational fear response. Moreover, we do not exclude the possibility that the reduced observational fear response in the picrotoxin+EtOH group was due to dysfunctional ACC neuronal circuit activity. Therefore, in this re-revised manuscript, we have toned down our claims about the direct relationship between the E/I balance and observational fear response (P11, L259). Nonetheless, it is still true that previous reports suggest that the E/I balance is one of the critical factors for social behaviors (Yizhar et al., *Nature*, 2011; Han et al., *Nature*, 2012; Han et al., *Neuron*, 2014), and these reports are at least apparently consistent with our interpretation that the E/I balance is involved in socially evoked fear memory recall. Thus, we believe that our interpretation will be

useful for showing one of the directions in considering the role of the E/I balance in sociality.

Reviewer #2

2-1) Please refer to and cite the intranasal oxytocin paper in the Introduction.

In addition to papers which we have already cited (i.e., Hurlemann et al., J. Neurosci., 2010; Krueger et al., Soc. Cogn. Affect. Neurosci., 2013; Abu-Akel et al., Soc. Neurosci., 2015; Shamay-Tsoory et al., Psychoneuroendocrinology, 2013; De Dreu et al., Science, 2010; De Dreu et al., Proc. Natl. Acad. Sci. USA, 2011), we have listed other articles about the effects of intranasal oxytocin on inferring other's emotion (Aoki et al., Brain, 2014) and sociocommunication (Watanabe et al., JAMA psychiatry, 2014) in autistic persons (P4, L76).

2-2) Note the duration of co-housing in the opening lines of the Results – this is a critical variable.

We agree with this comment. We have described the duration of co-housing in the Results section (P4, L93).

2-3) ‘neighboring compartment’ not neighboring room’ in the first paragraph of the Results.

According to this suggestion, we have replaced the term "neighboring room" with "neighboring compartment" in the Results section (P5, L98).

2-4) Please note where the hippocampal CA1 sampling was done – dorsal or ventral?

The samples were collected from the dorsal hippocampal area. We have added this

information in the manuscript (P8, L171).

2-5) Statistical results are sometimes presented in the Results and other times (mainly) the Figure Legends – please be consistent.

Thank you for this comment. For consistency, we have written all the detailed statistical values in the Figure Legends.

2-6) Add the lux level used in the open field and elevated plus-maze.

The lux level used in the open field test and elevated plus-maze test was 165 lux. We have added this information in the Methods section (P18, L417; P19, L431).

2-7) Note on page 20 when precisely mice were killed post testing.

Mice were killed 5 min after the fear observation. We have added this explanations in the Methods section (P20, L463).

2-8) For Figure 4 BLA CTb injections – show an example of the injection site expression.

We have added an example image of the CTB injection site in the BLA in Figure 4.

Reviewer #3

3-1) The revised manuscript by Dr Ikegaya and colleagues is greatly improved. The authors worked very hard to address my concerns, and the additional experiments, analyses, revamped statistics, and updated text make the paper better. I have only one remaining concern. The fact that ethanol does not enhance defensive immobility in the FO-only group does not support their

general assertion that ethanol enhances empathy-like behavior. The authors have attempted to explain this using the argument that there is a combined memory recall and fear transmission effect. I think this is an acceptable argument, but the manuscript language needs to be tightened throughout to reflect this interpretation. For example, the abstract states that ethanol, “facilitates observational fear transmission” (Line 32). This and other statements need to be changed to reflect their updated hypothesis so as to not be misleading. Possibilities could include using “socially evoked fear memory recall” or “priming-induced fear transmission”. If the authors update the text of the manuscript to reflect their updated hypothesis, I am happy to recommend publication.

Thank you for your precious opinion. In response to receiving a similar comment from Reviewer #1, we have decided to change our argument concerning "empathy" and have rewritten this manuscript into the context of "socially induced fear memory recall". At the same time, we have carefully fixed our inappropriate expressions, including the parts that you pointed out. We appreciate these comments, which lead us to finish this manuscript from a scientific point of view.

REVIEWERS' COMMENTS:

Reviewer #1 (Remarks to the Author):

The authors have revised the manuscript appropriately incorporating the reviewer's comments. I think that it is ready to be published now.